# Striatal dopamine can enhance both fast working memory, and slow reinforcement learning, while reducing implicit effort cost sensitivity

Andrew Westbrook [1] ✉, Ruben van den Bosch[2,3], Lieke Hofmans[4,9], Danae Papadopetraki[2,3], Jessica I. Määttä [5], Anne G. E. Collins [6], Michael J. Frank [7,8,10] & Roshan Cools [2,3,10]

Associations can be learned incrementally, via reinforcement learning (RL), or stored instantly in working memory (WM). While WM is fast, it is also capacity-limited and effortful. Striatal dopamine may promote WM, by facilitating WM updating and effort exertion and also RL, by boosting plasticity. Yet, prior studies have failed to distinguish between the effects of dopamine manipulations on RL versus WM. N = 100 participants completed a paradigm isolating these systems in a double-blind study measuring dopamine synthesis with [$^{18}$F]-FDOPA PET imaging and manipulating dopamine with methylphenidate and sulpiride. We find that learning is enhanced among high synthesis capacity individuals and by methylphenidate, but impaired by sulpiride. Methylphenidate also blunts implicit effort cost learning. Computational modeling reveals that individuals with higher dopamine synthesis capacity rely more on WM, while methylphenidate boosts their RL rates. The D2 receptor antagonist sulpiride reduces accuracy due to diminished WM involvement and faster WM decay. We conclude that dopamine enhances both slow RL, and fast WM, by promoting plasticity and reducing implicit effort sensitivity. This work was completed as part of a registered trial with the Overview of Medical Research in the Netherlands (NL-OMON43196).

Striatal dopamine signaling has been implicated in cortico-striatal plasticity[1–3] and reinforcement learning (RL) across species[2,4]. Yet, in humans, a substantial contribution to learning is mediated by working memory (WM) processes[5–12], and prior work linking individual differences in dopamine function to RL typically fails to disentangle the impact on incremental RL versus WM.

While WM is faster and more flexible, multiple constraints limit its contributions. Unlike RL, WM is capacity-limited and subject to

[1]Department of Psychiatry, Center for Advanced Human Brain Imaging Research, Rutgers University, Piscataway, NJ, USA. [2]Donders Institute for Brain, Cognition and Behaviour, Radboud University, Nijmegen, the Netherlands. [3]Department of Psychiatry, Radboud University Medical Center, Nijmegen, the Netherlands. [4]Department of Psychology, University of Amsterdam, Amsterdam, the Netherlands. [5]Department of Psychology, Stockholm University, Stockholm, Sweden. [6]Department of Psychology & Helen Wills Neuroscience Institute, University of California at Berkeley, Berkeley, CA, USA. [7]Cognitive, Linguistics, & Psychological Sciences Department, Brown University, Providence, RI, USA. [8]Carney Institute for Brain Science, Brown University, Providence, RI, USA. [9]Present address: Institut du Cerveau, Pitié-Salpêtrière Hospital, Paris, France. [10]These authors contributed equally: Michael J. Frank, Roshan Cools. ✉ e-mail: andrew.westbrook@rutgers.edu

decay[13–15]. WM may thus play a smaller role when there are more items to remember, and when they were encountered farther in the past. WM is also effort-costly, and people may forgo WM-based strategies to avoid the effort[16–19].

Importantly, theoretical models and empirical data implicate striatal dopamine signaling in both RL, by altering synaptic plasticity, and WM via formation and expression of cortico-striatal synapses that encode gating policies[20–23]. Furthermore, striatal dopamine can also make WM less effort-costly[24–28]. While WM is effortful, striatal dopamine signaling promotes willingness to exert effort by making people more sensitive to potential benefits, and less sensitive to effort costs[24]. If striatal dopamine signaling promotes reliance on WM—either by shaping corticostriatal synapses to facilitate gating, or by making WM less effortful—this can amplify effective learning rates, confounding inferences about the direct effects of striatal dopamine signaling on RL.

Here, we use a paradigm designed to distinguish between contributions of WM and RL to stimulus-response learning. The reinforcement learning working memory (RLWM) task[10] manipulates the degree to which people can rely on WM. Set sizes vary (between two and five items) across blocks of trials, thus taxing WM load, delay, and interference to varying degrees. While healthy adults *can* rely mostly on WM (rather than RL) when there are only two items to encode in recent experience, participants *must* rely increasingly on RL as set sizes grow to exceed WM capacity[7,9,10,29]. We furthermore include a surprise test phase, after learning, to probe the durability of RL-informed learning in terms of participants' ability to recall features of the stimuli after a long delay.

To study dopamine's effects on RL and WM, we employ a combination of methods. We isolate the effects of dopamine signaling in the striatum by measuring individual differences in the rate at which dopamine is synthesized in presynaptic striatal terminals using [18F]-FDOPA PET imaging. We also manipulate dopamine signaling, while participants perform the RLWM task in separate sessions, by administering a placebo, methylphenidate—a dopamine (and noradrenaline) reuptake blocker commonly used to treat attention deficit hyperactivity disorder, or sulpiride—a D2 receptor antagonist commonly used to treat psychosis in schizophrenia.

To anticipate our results, we find that striatal dopamine is related to enhanced performance on the RLWM task. Behavioral analyses and computational modeling reveal that higher dopamine synthesis capacity promotes faster learning by increasing reliance on WM. Sulpiride undermines performance specifically because it increases interference within WM when there is an increasing number of items between successive encounters of the same stimulus. Furthermore, we find that methylphenidate boosts the rate of RL, controlling for WM contributions to the learning process. Finally, we find that while WM is effort-costly, methylphenidate can blunt implicit effort cost learning associated with more demanding tasks.

## Results

In each of three drug sessions—placebo, methylphenidate, and sulpiride—participants completed the RLWM task, which was designed to dissociate RL and WM processes during stimulus-response learning[10]. On a given trial, participants learn to associate pictures with one of three buttons by trial-and-error (Fig. 1). Participants are given feedback about the accuracy of each response ($R_j$). To distinguish contributions of RL versus WM, stimuli are presented in blocks of varying set size. By this design, WM can dominate when there are only two items to remember, typically separated by minimal delays (e.g., one or two trials). Conversely, RL necessarily plays a larger role for larger sets sized blocks (up to five items in a block) as the number of items and delays (defined here as the number of trials since the last correct response for a given stimulus) grow to exceed WM capacity.

### Both reinforcement learning and working memory contribute to performance

Conjoint contributions of WM and RL are implied by the shape of learning curves by stimulus iteration (Fig. 2). Curves are steeper when set size is smaller and people could, in principle, rely more on WM. In a logistic regression of accuracy on set size and previous iteration count (the number of times a stimulus has been encountered), accuracy is higher with more iterations of each stimulus ($\beta = 1.90$; $p < 2.0 \times 10^{-16}$) and for smaller set sizes ($\beta = -0.30$; $p < 3.6 \times 10^{-10}$). An interaction between these factors implies that effective learning rates are larger in smaller set size blocks ($\beta = -0.11$; $p = 0.0016$). Prior computational and neurophysiological evidence[7,9,10,29,30] supports the hypothesis that steeper learning curves for smaller set sizes reflect greater reliance on WM rather than faster RL.

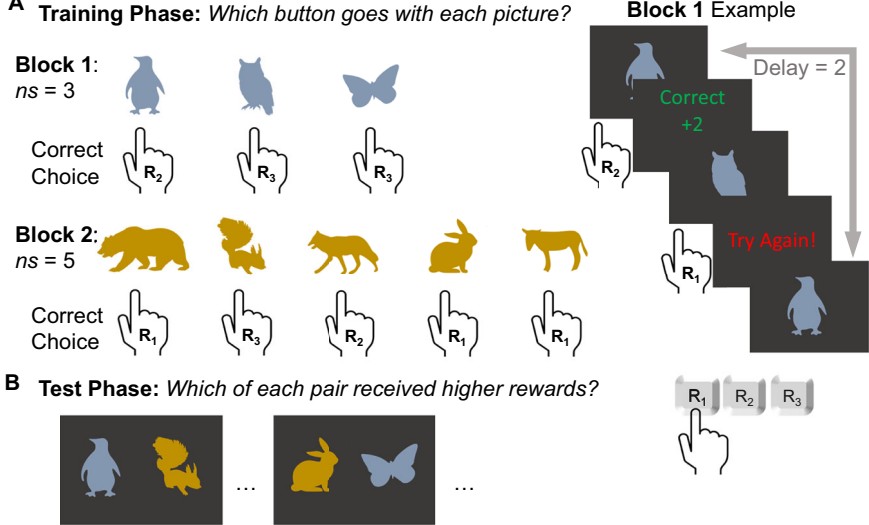

**Fig. 1 | RLWM task schematic. A** Training Phase: in each block participants are shown sets of images with set size (*ns*) varying by block (between *ns* = 2 and *ns* = 5). Participants learn through trial and error which of three buttons to press in response ($R_j$) to each stimulus. If they respond correctly, they are rewarded points (+2 or +1, amounts determined probabilistically, see "Methods" for full details) and if they are incorrect, they receive no points. **B** Test phase: participants are shown pairs of stimuli selected from across all training phase blocks and instructed to select the item in each pair for which they recall receiving the most rewards.

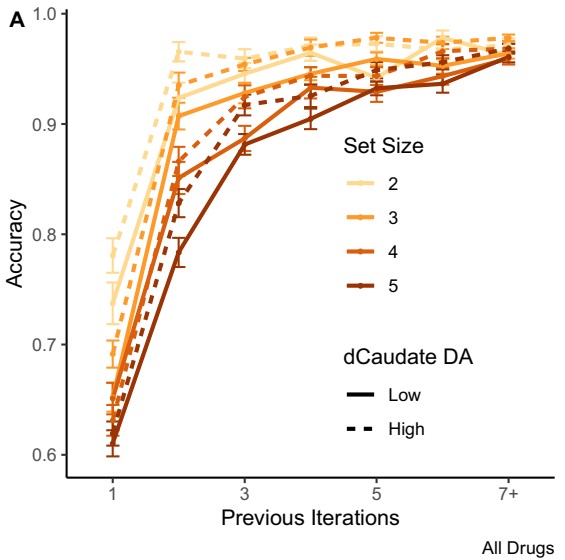
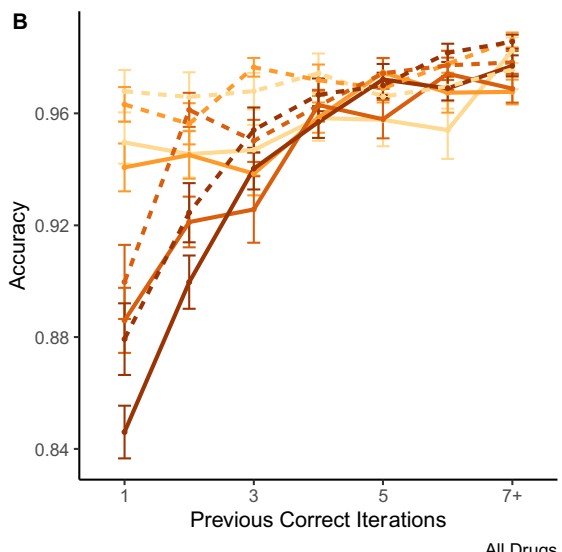

**Fig. 2 | Learning phase performance varies by dopamine synthesis capacity and set size.** Accuracy as a function of set size, individual differences (median split, $n = 46$ in each group) in dopamine synthesis capacity in the dorsal caudate nucleus (dCaudate DA), and **A** the number of previous iterations for each stimulus or **B** the number of previous correct iterations for each stimulus. Accuracy increases with more iterations of each stimulus on all drugs, and it is higher for those with higher dopamine synthesis capacity. Error bars indicate ± SEM.

It has further been demonstrated in both behavior and neural activity that people rely more on WM for novel items when fast and flexible updating offers the best performance benefits, and they rely more on RL for familiar items when slow but robust RL-cached values have had the chance to form[9,10,29,30]. An analysis of performance early versus late in a block supports this distinction, with WM appearing to contribute more to performance early, while RL contributes more late in a block. Two hallmarks of WM reliance—sensitivity to set size (Fig. 3A) and delay (Fig. 3B)—are apparent early in a block (fewer than three previous correct iterations for each stimulus) and are minimal late in a block (the last two iterations for each stimulus) across all three drug sessions (Fig. 3). Two-way ANOVAs reveal that participants' average performance across all three sessions varies as a function of set size ($F(1732) = 127$, $p < 2.0 \times 10^{-16}$), early versus late in a block ($F(1732) = 7.28 \times 10^3$, $p < 2.0 \times 10^{-16}$) and their interaction ($F(1732) = 104$, $p < 2.0 \times 10^{-16}$), and also effects of delay ($F(1,2388) = 596$, $p < 2.0 \times 10^{-16}$), early versus late ($F(1,2388) = 102$, $p < 2.0 \times 10^{-16}$), and their interaction ($F(1,2388) = 30.5$, $p = 3.6 \times 10^{-8}$). Thus, these data support the hypothesis that when people encounter novel items, they rely heavily on WM, and thus also show limitations due to capacity and interference within WM, but they shift to RL as the number of previous correct iterations grows and RL-based information becomes more reliable. This inference is also consistent with evidence that although people perform better for smaller set sizes early in a block, neural indices of RL grow across a block and do so faster in higher set-sized blocks, when participants rely more on RL versus WM[30].

**Striatal dopamine variously enhances performance**

Striatal dopamine signaling improves performance in multiple ways. We employed complementary tools to dissect these effects. First, we measured individual differences in dopamine synthesis capacity using [18F]-FDOPA PET imaging. Dopamine synthesis capacity is correlated across five striatal sub-regions (ICC = 0.75; $p = 3.1 \times 10^{-4}$), yet we focused our analysis on the dorsal caudate nucleus, where dopamine signaling has previously been implicated in RL about higher cognition[31–34], WM gating[20,23,26,35], and cost-benefit decision-making about cognitive effort[24,36–38]. We also had participants complete the task in three drug sessions, after taking methylphenidate—a dopamine and noradrenaline transporter blocker which should amplify striatal

dopamine signaling, and sulpiride—a selective D2 receptor antagonist, or placebo.

To evaluate the effect of dopamine on performance, we fit a hierarchical Bayesian model regressing trial-wise accuracy on drug, individual differences in dopamine synthesis capacity, and session number. In our model we also simultaneously estimate the effects of set size, the number of previous correct iterations (the number of prior trials on which a correct response was given for each stimulus), and the delay (number of trials) since the last correct iteration, along with higher order interactions with drug status and dopamine synthesis capacity (see Supplementary Table S1 for the full results). The fitted model reveals that striatal dopamine signaling clearly enhances performance. Specifically, higher dopamine synthesis capacity ($\beta = .17$; $p = .026$) and methylphenidate versus placebo ($\beta = 0.41$; $p = 1.6 \times 10^{-6}$) both increase accuracy, while sulpiride decreases accuracy ($\beta = -0.27$; $p = 1.7 \times 10^{-4}$; Fig. 3).

To understand why striatal dopamine synthesis capacity and methylphenidate boosted performance and sulpiride undermined it, we fit an RL model to behavior (adapted from refs. 10,12), examining how people learn to select the correct action for each stimulus in each block. The algorithm combines a WM component featuring instantaneous learning, capacity limits, and susceptibility to decay, and a capacity-unlimited RL component with incremental learning rates (see Methods for full details).

Fitted model parameters imply two ways in which striatal dopamine signaling boosts performance. First, it boosts performance by increasing the likelihood that people rely on WM ($\rho$; Fig. 4A, B), which facilitates fast and flexible acquisition of new associations. Second, striatal dopamine signaling also appears to increase the learning rate in the RL system ($\alpha_{RL}$; Fig. 4C), which describes the rate at which incrementally acquired stimulus-response associations can contribute to action selection.

A hierarchical regression of the parameter $\rho$ (WM reliance) on dorsal caudate dopamine synthesis capacity and drug, controlling for session number, reveals a positive effect of dopamine synthesis capacity on placebo ($\beta = 0.23$; $p = 0.029$; Fig. 4A), no effect of methylphenidate vs placebo ($\beta = 0.083$; $p = 0.48$), and a negative effect of sulpiride vs placebo ($\beta = -0.35$; $p = 0.0028$; Fig. 4B; Supplementary Table S3) on WM reliance. We note that we separately regressed

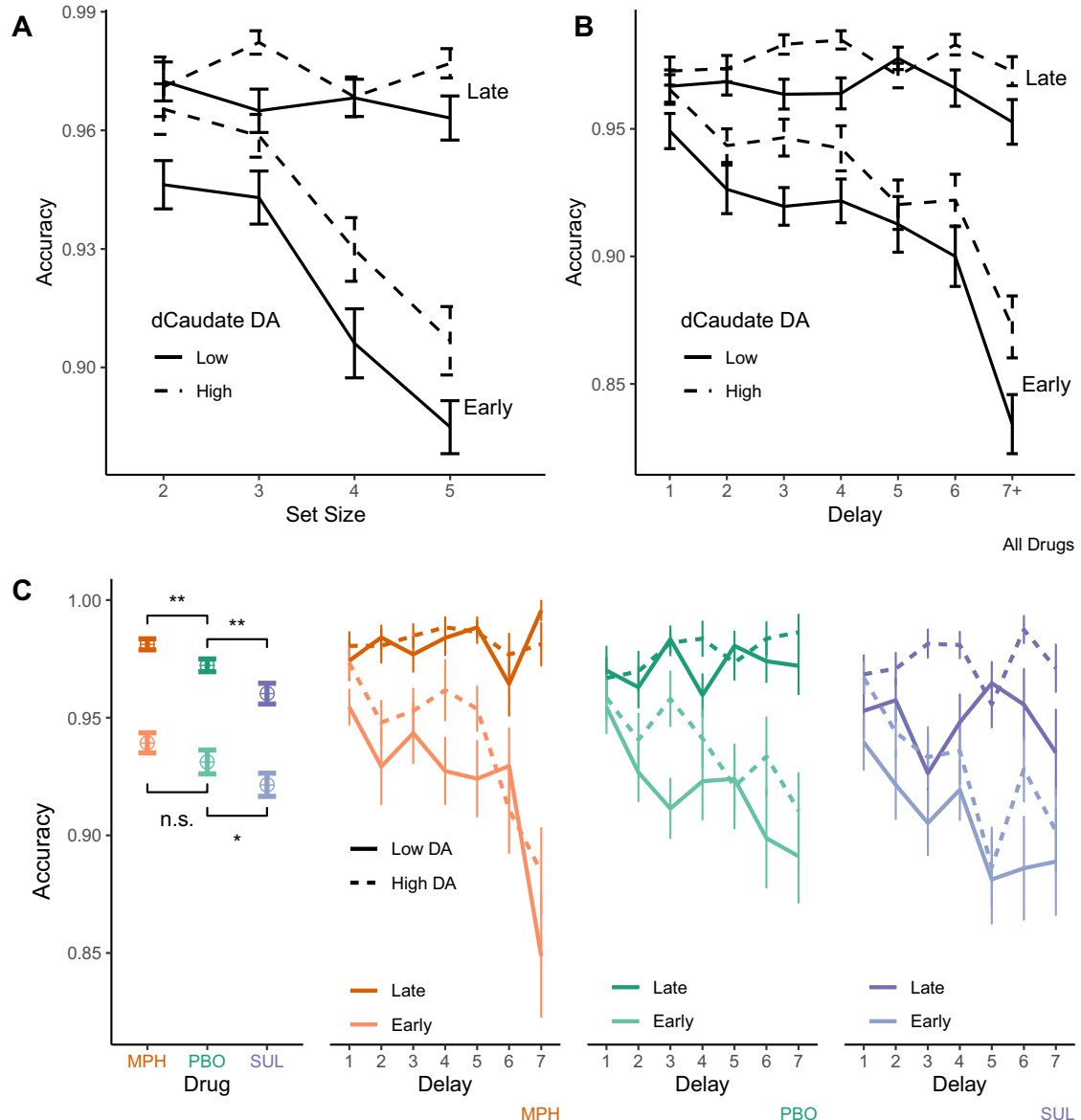

**Fig. 3 | Accuracy as a function of set size, delay, drug, and dopamine synthesis capacity.** Accuracy varies by dopamine synthesis capacity in the dorsal caudate nucleus (dCaudate DA) across all three drug sessions: methylphenidate (MPH; $n = 45$ and 46 for high and low dCaudate DA, respectively—referred to as "High DA" and "Low DA" in the figure), placebo (PBO; $n = 46$, 45), and sulpiride (SUL; $n = 46$ and 46), early (the first two iterations of all stimuli; $n = 46$ and 46) versus late (the last two iterations; $n = 46$ and 46) in each block. The effects of **A** set size and **B** delay are evident early, but not late in each block. **C** Means and standard errors reflect that methylphenidate significantly increases, and sulpiride decreases accuracy

relative to placebo, both early and late in a block (albeit the increase in early performance on methylphenidate is non-significant). Further analyses reveal that methylphenidate boosts performance in late versus early trials to a greater extent than placebo in separate mixed effect logistic regressions of accuracy, early versus late in a block. \*,\*\* indicate $p < 0.05$ and $p < 0.01$; main effects of the early model, reveal that the effect of sulpiride versus placebo is significant (in two-sided z-tests at $p = 0.026$) and the effect of methylphenidate versus placebo is not ($p = 0.20$). In the late trials, both effects are significant ($p = 0.0078$ and $p = 0.0039$, respectively). Error bars indicate ± SEM.

multiple model parameters onto dopamine factors and this dopamine synthesis capacity effect does not survive when we correct for these multiple comparisons ($p_{Bonferroni} = 0.087$; Supplementary Tables S3–S5). Nevertheless, a two-way ANOVA, correcting for multiple comparisons, confirms a main effect of dopamine synthesis capacity ($F(1251) = 8.94$, $p_{Bonferroni} = 0.0093$) across sessions and a main effect of drug across participants ($F(2251) = 4.44$, $p_{Bonferroni} = 0.039$), but no dopamine synthesis capacity by drug interaction ($F(2251) = 2.06$, $p_{Bonferroni} = 0.39$). Collectively, these results support the hypotheses that people who can synthesize dopamine at a higher rate rely more on WM in general and that sulpiride reduced WM reliance.

The hypothesis that participants with higher dopamine synthesis capacity rely more on WM is further supported by evidence that they tend to perform better early in a block when items are novel and WM affords better performance (Fig. 3A, B). A hierarchical logistic regression, restricted to early trials, regressing accuracy on set size, drug, and dopamine synthesis capacity, reveals that participants with higher dopamine synthesis capacity perform better when WM plays a bigger role ($\beta = 0.14$; $p = 0.019$; Supplementary Table S6). Although the same effect is not significant in late trials ($\beta = 0.12$; $p = 0.18$; Supplementary Table S6), the data do not support a distinction between early and late trials. The two-way interaction between dopamine synthesis capacity

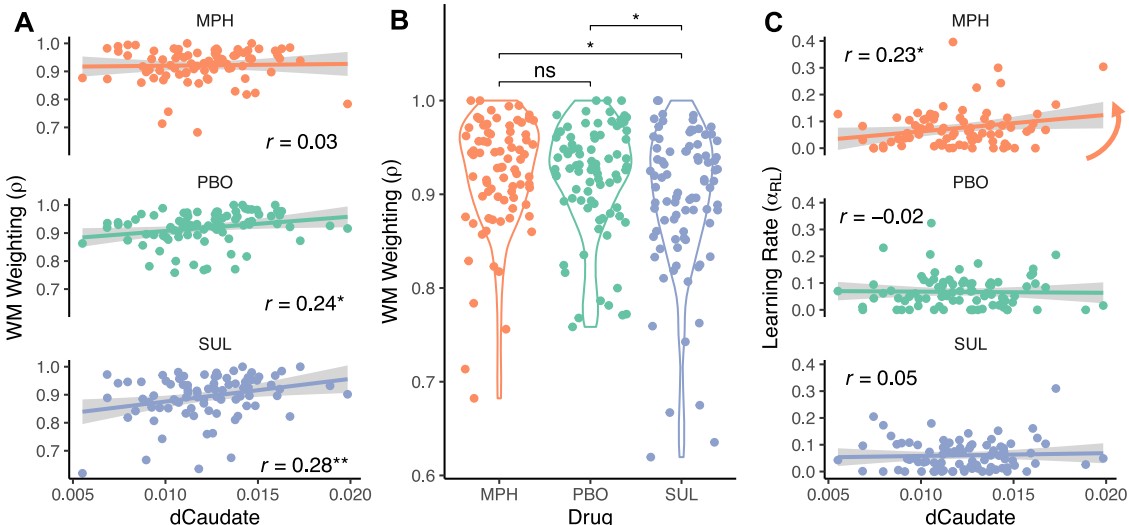

**Fig. 4 | Model parameters vary by dopamine synthesis capacity and drug.**
**A**, **B** WM weighting ($\rho$) and **C** the RL learning rate ($\alpha_{RL}$) parameters from reinforcement learning algorithm as a function of individual differences in the dopamine synthesis capacity in the dorsal caudate nucleus of the striatum (dCaudate) across three drug sessions: MPH methylphenidate, PBO placebo, and SUL sulpiride. Drug effects and Pearson's correlation values are reported along with their significance level: *,** indicate $p < .05$ and $p < .01$. For **B** two-sided, paired $t$-tests reveal significant differences between WM weighting on PBO versus SUL ($t(74) = 2.68$, $p_{Bonferroni} = 0.018$) and MPH versus SUL ($t(74) = 2.87$, $p_{Bonferroni} = 0.016$), but not MPH versus PBO ($t(74) = 0.89$, $p = 0.38$). Colored lines are a linear fit to the data and gray bands indicate 95% CI.

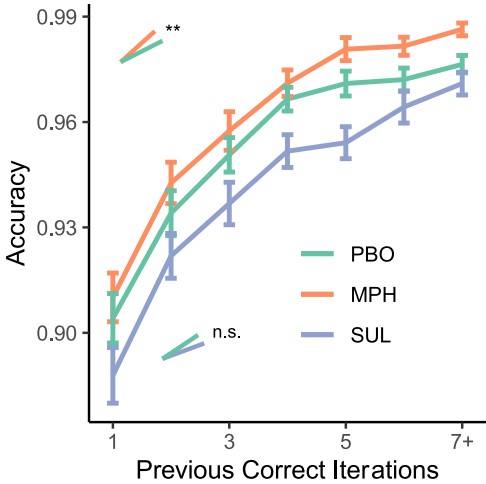

**Fig. 5 | Accuracy as a function of previous correct iterations for each stimulus and drug.** The effect of previous correct iteration is larger on methylphenidate (MPH) versus placebo (PBO), and no different on sulpiride (SUL). The effect of MPH is significant: ** indicates $p < 0.01$ (in two-sided $z$-test at $p = 0.0042$). Error bars indicate $\pm$ SEM ($n = 91$ in each session).

and early versus late trials is not significant ($\beta = -0.0099$; $p = 0.89$; Supplementary Table S8).

There is also evidence linking dopamine synthesis capacity to WM reliance in the full logistic regression (including all trials and task variables; Supplementary Table S1). Namely, participants with higher dopamine synthesis capacity have a larger effect of set size on accuracy ($\beta = -0.10$; $p = 0.029$). Additionally, a two-way interaction indicating stronger delay effects in larger set sized blocks ($\beta = -0.28$; $p = 2.8 \times 10^{-14}$) is also larger for participants with a higher dopamine synthesis capacity ($\beta = -0.08$; $p = 0.016$). Thus, model-independent analyses converge on the hypothesis that people who can synthesize dopamine at a higher rate rely more on WM, thus boosting their performance on early trials, and also making them more susceptible to set size and delay effects on trial-wise accuracy.

Even when controlling for the effect of striatal dopamine signaling on WM, however, there is evidence that dopamine boosts incremental RL learning rates, as noted above (Fig. 4C). A hierarchical regression of the learning rate parameter $\alpha_{RL}$ on dopamine, controlling for session number, reveals a two-way interaction such that methylphenidate boosts learning rates for those with higher dopamine synthesis capacity ($\beta = 0.30$; $p = 0.032$; albeit this effect does not survive correction for multiple comparisons across model parameters $p_{Bonferroni} = 0.096$; Supplementary Tables S3–S5). There were no main effects of dopamine synthesis capacity ($\beta = -0.027$; $p = 0.79$), methylphenidate ($\beta = 0.15$; $p = 0.29$), or sulpiride ($\beta = -0.10$; $p = 0.46$). However, the interaction reflected a positive correlation between dopamine synthesis capacity and $\alpha_{RL}$ during the methylphenidate session ($r = 0.23$; $p = 0.042$) that was absent in the placebo session ($r = -0.020$; $p = 0.86$). These results support the hypothesis that by blocking dopamine reuptake and amplifying post-synaptic signaling, methylphenidate increases the rate of RL, especially for those who synthesize dopamine at a higher rate.

The inference that methylphenidate boosts learning rates is also supported by model-independent analyses. Specifically, the full logistic regression of accuracy on dopamine and task variables (Supplementary Table S1) reveals that methylphenidate increases accuracy ($\beta = 0.41$; $p = 1.6 \times 10^{-6}$) and amplifies the effect of previous correct iterations (two-way interaction: $\beta = 0.20$; $p = 3.7 \times 10^{-3}$; Fig. 5)—a basic index of RL processes[9,30,39]. Although dopamine synthesis capacity also increases accuracy overall (noted above), it does not, in contrast with methylphenidate, interact with previous correct iterations ($\beta = -0.01$; $p = 0.84$). Thus, on methylphenidate, people perform better in part because their incremental improvement with each rewarded trial is greater. Consequently, the overall improvement in accuracy between early and late trials is larger on methylphenidate versus placebo. A mixed-effects logistic regression of accuracy restricted to just early and late trials indeed reveals that the improvement between early and late trials is larger on methylphenidate versus placebo (two-way interaction: $\beta = 0.29$; $p = 0.030$; Supplementary Table S8). These results converge on the hypothesis that methylphenidate accelerates RL, leading to faster incremental acquisition of stimulus-response contingencies and a bigger overall improvement from early to late trials in each block.

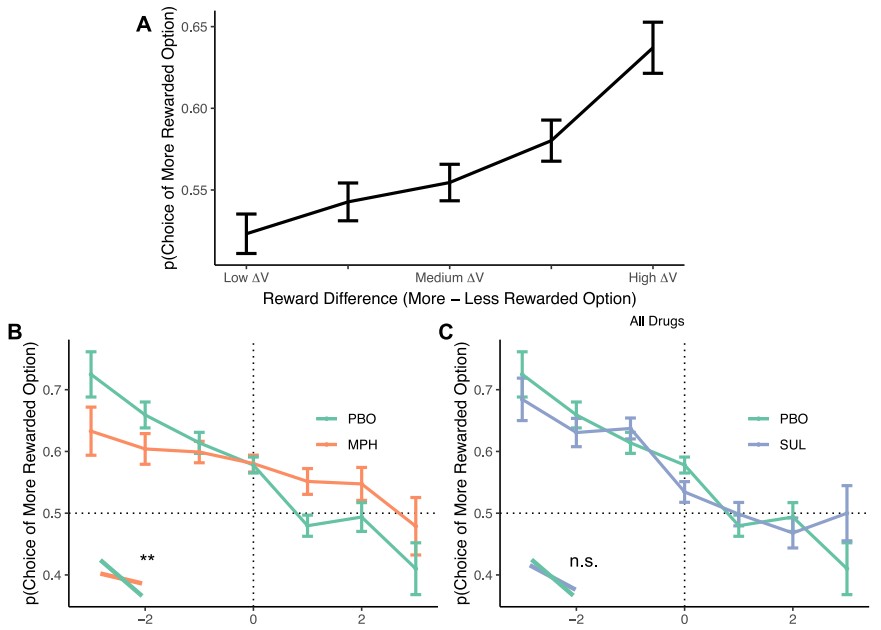

**Fig. 6 | Test phase performance depends on actual rewards received and drug.** Performance (selection of the option rewarded at a higher rate) as a function of the differences in (**A**) actual reward statistics (Δ*V*) and (**B**, **C**) the set size of the block from which the stimulus was drawn. **A** Participants successfully identify the more highly rewarded outcome higher than chance, and increasingly so as the difference in actual reward rates increases. **B**, **C** Participants are less likely to indicate an option was more highly rewarded if it came from a larger set-sized block. Relative to placebo (PBO), this set size difference effect is smaller on methylphenidate (MPH) and no different on sulpiride (SUL). Error bars indicate ± SEM (PBO: $n = 78$, MPH: $n = 69$, SUL: $n = 80$).

Sulpiride, in contrast, undermines performance. According to mixed-effects logistic regressions of accuracy, restricted to either early or late trials, this is true both early ($\beta = -0.20$; $p = 0.024$; Supplementary Table S6) and late in a block ($\beta = -0.35$; $p = 0.0072$; Supplementary Table S7). One reason that people performed worse on sulpiride appears to be that they rely less on WM, in general. Indeed, fitted parameters from the RL algorithm reveal that sulpiride had a negative main effect on the WM reliance term $\rho$ ($\beta = -0.35$; $p = 0.0028$; Supplementary Table S3), as noted above.

There is evidence that people relied less on WM overall, after taking sulpiride, because WM itself was less reliable. In the full logistic regression of accuracy on dopamine and task variables (Supplementary Table S1), the negative effect of sulpiride on accuracy, was accompanied by a two-way interaction, indicating that delay effects were stronger on sulpiride versus placebo ($\beta = -0.11$; $p = 0.047$). Thus, participants tend to perform worse when there is a bigger delay since the last previous correct iteration for a given stimulus, to a greater extent on sulpiride versus placebo. Thus, sulpiride may have reduced performance, in part, because it amplifies delay effects, undermining the contributions of WM to the learning process.

In sum, parameters from our RL model converge with our model-independent analyses on the interpretation that greater striatal dopamine signaling among those who synthesize dopamine at a higher rate predicts greater reliance on WM. Conversely, on sulpiride, performance decreases and this appears to reflect less reliance on WM across participants. Our results furthermore indicate that methylphenidate increases the rate of RL, especially among those who synthesize dopamine at a higher rate, even after controlling for the drug's effects on WM.

**Methylphenidate blunts implicit effort cost sensitivity**
Following stimulus-response learning, participants were presented with a surprise test phase in which they are asked to select which of two stimuli received greater rewards (Fig. 1). Stimuli were selected pseudo-randomly from those previously encountered across all training phase blocks. The intent of this test phase was to study RL-based value

representations—which should be robust to the decay across blocks—after learning. Following prior work[7,30,39], we examine whether the reward statistics people learn through RL are influenced by the degree to which people discount rewards by the cognitive effort they exerted when rewards were received.

To evaluate the effects of task and dopamine variables on RL-based value representations, we fit a hierarchical Bayesian logistic regression of pairwise accuracy (correctly identifying the stimulus that received higher rewards) on task variables, drug, and dopamine synthesis capacity (see Supplementary Table S2 for full results). Replicating prior work[39] we find that participants faithfully track reward statistics, correctly picking the stimulus which was rewarded at a higher rate (intercept: $\beta = 0.08$; $p = 0.0090$), and increasingly so as the difference in actual rewards increased ($\beta = 0.31$; $p = 4.2 \times 10^{-15}$; Fig. 6A).

Also replicating prior work[39], we find that cached reward values are perceived as less rewarding when they came from larger set-size blocks—an effect previously interpreted as reflecting implicit sensitivity to effort costs. Specifically, controlling for the rewards people received for each stimulus during the learning phase, participants assign a lower reward value to stimuli that had been encountered in larger set-size blocks. This effect is captured by a negative effect of the difference in set sizes between the stimulus which objectively received more rewards and the stimulus that received fewer rewards ($\beta = -0.28$; $p = 7.1 \times 10^{-8}$; Fig. 6B: collapsed across differences in actual reward rates). That is, while participants track the rewards associated with each stimulus, they treat rewards received in the context of higher set sizes, and thus higher WM demands, as subjectively less rewarding. It also converges with our own prior work in which we find that participants treat higher WM demands as effort costly, requiring greater reward offers to offset these costs[18,24].

Importantly, we also find evidence that striatal dopamine blunts this implicit effort discounting effect. Specifically, the effect of set size on perceived rewards was significantly less on methylphenidate versus placebo ($\beta = 0.12$; $p = 0.025$; Fig. 6B). This effect of set size on perceived rewards was not influenced by dopamine synthesis capacity ($\beta = -0.01$; $p = 0.85$). This effect also does not reflect differences in the

ability of participants to track reward statistics between drug sessions. Indeed, test phase accuracy (the rate at which participants correctly identify the more rewarded stimulus) is no different on methylphenidate versus placebo, and there is also no drug by reward rate difference interaction (both $p$'s > 0.57; Supplementary Fig. S6). In a different experiment conducted within the current study, we find that striatal dopamine signaling increases sensitivity to reward benefits and decreases sensitivity to effort costs during decision-making about cognitive effort[24]. This result converges with the present finding that dopamine signaling can blunt implicit effort sensitivity during reward learning. Thus, by amplifying striatal dopamine signaling, methylphenidate not only increases sensitivity to benefits versus costs during action selection[24], but may also alter how people learn about effort costs and benefits in the first place.

## Discussion

Striatal dopamine promotes corticostriatal plasticity and thereby facilitates RL[1–4]. Numerous studies have attempted to link stronger dopamine signaling capacity to faster RL. However, this exercise is complicated by the fact that striatal dopamine may also govern the degree to which people rely on fast and flexible WM to accomplish the same tasks[10]. Potential effects on both RL and WM implies that it is essential to control for either factor to determine the extent to which dopamine affects the other. This is true whether the goal is to evaluate the true rate of RL, or the degree to which WM contributes to learning beyond RL processes.

In this study, we employ a task that dissociates the relative contributions of RL and WM systems to learning and examine how dopamine influences each. We find that striatal dopamine signaling—modulated by either dopamine synthesis capacity or dopamine drugs—can promote learning, with distinct effects on both RL and WM. Specifically, higher dopamine synthesis capacity in the dorsal caudate nucleus predicts greater reliance on WM, while antagonism of dopamine receptors with sulpiride reduces performance by reducing reliance on WM. By blocking dopamine reuptake and thereby amplifying dopamine signaling, methylphenidate also boosts performance in interactions with striatal dopamine synthesis capacity. Specifically, methylphenidate increases RL learning rates for people who synthesize dopamine at a higher rate. Importantly, we find that dopamine promotes both WM and RL processes when controlling for the effects of dopamine on the other. Thus, we infer that striatal dopamine signaling enhances learning, both fast (via increased reliance on fast and flexible WM) and slow (by boosting relatively slow, but practically unlimited capacity RL).

We speculate that reliance on WM correlates with striatal dopamine synthesis capacity because the latter may shape a trait policy to rely on WM in general, in novel contexts. This could help explain why dopamine synthesis capacity may correlate with individual differences in WM capacity[26,40]. Correlations between WM capacity and dopamine synthesis capacity may be obtained if study and task designs are sensitive to individual differences in the degree to which people rely on WM (although they will not always obtain)[41]. But why does dopamine synthesis capacity promote reliance on WM? One idea is related to the emerging hypothesis that WM is effort-costly and stronger striatal dopamine signaling helps overcome those effort costs[24,27,38,42–52]. In a prior study, we showed that people are more willing to accept offers to perform more demanding WM tasks for money if they have higher dopamine synthesis capacity and on methylphenidate versus placebo[24]. Our interpretation was that striatal dopamine signaling influences both the learning and expression of cost-benefit policies governing WM allocation—making people more sensitive to performance benefits and less sensitive to effort costs.

The present findings constitute an important complement to that conclusion. Namely, while the prior study primarily examined prospective decisions about WM tasks, after effort costs had already been learned, the present study reveals that dopamine may shape value learning about effort costs themselves. Specifically, we find that people treat rewards earned in the context of higher WM demands as subjectively less rewarding—an effect which we interpret as implicit discounting of rewards by increasing cognitive effort costs. Critically, we find that methylphenidate blunts this implicit effort-discounting effect. Thus, stronger striatal dopamine signaling may make people both more willing to exert effort for tasks they have experienced in the past and experience the tasks as less costly when they learn about them in the first place (cf. [50]). An important caveat is that, unlike prior work showing that methylphenidate can alter explicit cost-benefit decision-making, our measure of effort costs is implicit and the inference that dopamine modulates cost learning during task performance is indirect. Taken together with our prior study, our present results support the hypotheses that striatal dopamine signaling can make WM tasks less effort costly, both as people learn about the costs of performing the task, and at the moment of choice, when they decide whether to expend effort in the future. We note that these effects of dopamine on prospective decision-making and effort cost learning may complement the proposed role of norepinephrine in energizing on-going effort expenditure[53].

Many forms of psychopathology have been associated with aberrant rates of RL. However, our results confirm prior work (e.g.[8–10]) showing that it is crucial to control for the degree to which WM contributes to the learning process when trying to estimate learning rates for an RL system. Without a WM component in our learning algorithm, the learning rates for the RL system would have to be at least an order of magnitude larger to capture the rapid rates at which people acquire stimulus-response associations (in most cases one or two trials; cf. fig. [2]B). Moreover, without accounting for WM contributions to the learning process, the learning rates in the RL system would have to vary by set size. Thus, both within- (e.g., set size) and between-subjects factors (e.g., WM reliance, $\rho$) would confound estimates of incremental learning rates in an RL system if WM is not accounted for. This result highlights the prospect that many prior studies showing psychopathological or neurological deficits in RL may have instead found evidence of deficits in WM.

Controlling for the contributions of WM to the learning process, we also find evidence that striatal dopamine signaling accelerates RL. Specifically, we find that methylphenidate boosts RL rates the most for people who synthesize dopamine faster in the dorsal caudate nucleus. This result mirrors prior work showing that methylphenidate promotes plasticity for RL about rewards, especially for people with high WM capacity[54–56]—a proxy for dopamine synthesis capacity[40]. Moreover, in another experiment conducted in the context of the current study, we find that methylphenidate increases accuracy and prefrontal BOLD signal response to rewards versus punishments, in a reversal learning task, to a greater extent among individuals with higher dopamine synthesis capacity[57]. One explanation for these effects is that methylphenidate acts by blocking reuptake, and so it will have an even larger effect for people who synthesize and therefore release more dopamine in response to reward in the first place. The combination of greater release and reuptake blockade causes dopamine to linger in the synapse for longer[58], thereby constructively amplifying LTP in response to phasic dopamine learning signals. It is also conceivable that dopamine synthesis capacity might boost RL learning rates as a main effect, but our study design was insensitive to this relationship. Future work could explore a wider range of set sizes (e.g., set size 6) where performance would depend to an even greater extent on RL mechanisms.

Although sulpiride clearly undermines performance, the mechanisms are somewhat less resolved. Sulpiride causes performance to decline both early and late in blocks, suggesting it may impact both RL and WM processes. Model parameters suggest that sulpiride diminishes performance because it reduces the degree to

which WM contributes to behavior. Our analyses further suggest that WM contributes less because of a faster decay of WM contents. Prior work has shown that WM and RL systems work cooperatively to accomplish learning, but, paradoxically, higher fidelity WM contents attenuate RL because better WM-based predictions result in smaller prediction errors[7,9]. Thus, if anything, faster-decaying, degraded WM representations should yield stronger RL and better performance late in a block. The fact that performance is also worse late in blocks suggests that sulpiride may have detrimental effects on both RL and WM systems.

While prior studies have shown a role for D2 receptors in supporting WM function[59–61], we did not predict that antagonism of D2 receptors with sulpiride would necessarily undermine performance or reduce reliance on WM. In fact, sulpiride may strengthen rather than weaken post-synaptic dopamine signaling by binding to pre-synaptic D2 autoreceptors, thereby releasing a break on dopamine release. In our own prior study[24], we found evidence that was consistent with the hypothesis that this dose of sulpiride strengthens post-synaptic dopamine signaling. In that study, we had independent measures (e.g., increased saccadic vigor) converging on the hypothesis that sulpiride increased postsynaptic dopamine signaling—though we are unable to make strong conclusions here. If the drug does have the same postsynaptic effects in the context of this task, we speculate that sulpiride may produce steeper effective WM decay by lowering the barrier for WM gating. If the barrier is sufficiently low, hyper-flexibility would undermine the stability of task-relevant representations across trials. Indeed, in prior studies, we have found that both the dopamine precursor levodopa[62] and the D2 agonist bromocriptine[63] can increase distractor vulnerability in cognitive control and WM tasks. Pre-synaptic effects in our prior study helped explain why sulpiride increased willingness to perform more demanding WM tasks by making people less sensitive to effort costs in our prior study. Thus, perhaps sulpiride, when binding pre-synaptically, may both reduce sensitivity to effort costs and make WM less effective by amplifying decay effects.

Finally, we note that while our paradigm is designed to dissociate the effects of dopamine on RL versus WM, we do not believe that these systems are independent. On the contrary, prior work has demonstrated that WM and RL systems interact cooperatively during learning such that predictions maintained in WM can inform, and therefore reduce the magnitude of prediction errors, paradoxically slowing the effective RL rate[9,29]. We do not, therefore, infer that the apparently selective effects of methylphenidate on RL and dopamine synthesis capacity on WM in our dataset imply a lack of crosstalk. It is conceivable, for example, that a real effect of dopamine synthesis capacity on RL might have been masked by a countermanding effect of dopamine synthesis capacity increasing WM reliance, which, in turn, blunted prediction errors. Similarly, crosstalk between the systems might have masked real effects of methylphenidate on WM contributions to learning or real effects of sulpiride on the RL system.

Our results support multiple, complementary mechanisms by which striatal dopamine influences task learning. Namely, we find that individual differences in dopamine synthesis capacity correlate positively with baseline propensity to rely on WM. We also find that sulpiride reduces reliance on WM, perhaps by undermining the stability of WM contents over time. These effects indicate that striatal dopamine can increase reliance on fast and flexible WM for learning. Yet, even after accounting for the effects of dopamine on WM, we find that striatal dopamine can accelerate slow learning processes as well. Namely, we find that methylphenidate accelerates RL rates, consistent with the hypothesis that striatal dopamine promotes plasticity on slower time scales, too. Finally, we find evidence that pharmacological enhancement of striatal dopamine signaling can blunt implicit effort cost learning that happens when people perform demanding tasks. This result complements prior work by showing that dopamine not only influences effort-based decision-making at the time of choice, but also shapes the how people learn about effort costs in the first place.

## Methods

The study was conducted in compliance with a protocol approved by the regional research ethics committee (Commissie Mensgebonden Onderzoek, region Arnhem-Nijmegen; 2016/2646; ABR: NL57538.091.16). 100 Healthy, young adult participants (ages 18–43, 50 men) were recruited from Nijmegen, The Netherlands to participate in a within-subject, double-blind, placebo-controlled study. Participants were screened to ensure that they are right-handed, Dutch-native speakers, healthy, neurologically normal, and without a history of mental illness or substance abuse. All participants gave written informed consent to participate in this study. The first and last participants were enrolled, respectively, on 10 February 2017 and 28 June 2018.

This experiment is part of a broader trial, investigating the effects of dopaminergic drugs on cognitive control, registered with the Overview of Medical Research in the Netherlands: https://onderzoekmetmensen.nl/en/trial/43196. This study complies with the International Committee of Medical Journal Editors (ICMJE) guidelines on reporting. Not all of the primary outcomes of the original registered trial are reported here. In this study, we report specifically on task behavior associated with the RLWM task, and also PET imaging of striatal dopamine synthesis capacity (described below). No conclusions regarding other outcomes of the registered trial are being made in this paper. A full characterization of our participant pool, a detailed list of exclusion criteria, intake procedures, full drug administration protocol, and methods for measuring dopamine synthesis capacity as well as the full set of covariate measures and tasks for this broader study, are detailed in Määttä et al.[64].

### General procedure and tasks

Participants completed five visits as part of a broader study of the effects of dopamine on cognitive control: one screening session, three pharmaco-imaging sessions with multiple tasks performed in and out of the fMRI scanner after being administered placebo, sulpiride, or methylphenidate, and a final PET session for measuring dopamine synthesis capacity. Participants were assigned to complete all three pharmaco-imaging visits in counterbalanced session order, however, drug session order was imperfectly counterbalanced. Consequently, 23, 15, and 10 participants took placebo on session number 1, 2, and 3, respectively, while the numbers were 12, 18, and 18 for sulpiride, and 13, 15, and 20 for methylphenidate. Given data loss and imperfect counterbalancing of drug by session order, we confirmed all inferences via hierarchical regression analyses, controlling for session order.

During screening, after providing written consent, participants completed—among other tests—medical and psychiatric screening interviews, as well as tests of WM capacity, and fluid intelligence.

Participants were asked to refrain from smoking or drinking stimulant-containing beverages the 24 h before a pharmaco-imaging session, and refrain from using psychotropic medication and recreational drugs 72 h before each session and cannabis throughout the experiment. At the beginning of a session, we measured baseline subjective measures, mood and affect, as well as temperature, heart rate, and blood pressure at baseline (also recorded after drug administration). Other tasks completed by participants, the results of which have been reported elsewhere, included tasks assessing sensitivity to cognitive effort costs and benefits[24,27], tasks measuring creativity[65,66], and a Pavlovian-to-instrumental transfer task[67]. Participants also completed two tasks in the fMRI scanner: one measuring striatal responsivity to reward cues and a reversal learning task[57]. Finally, after the behavioral sessions, but before the PET session, we also collected measures of

depression, state affect, BIS/BAS, impulsivity, and the degree to which participants pursue cognitively demanding activities in their daily life.

Participants were administered drugs prior to the task. To accomplish double-dummy blinding, participants took one drug capsule at each of two different time points: the first was either placebo or 400 mg sulpiride, while the second was either placebo or 20 mg methylphenidate. 160 min after taking methylphenidate or placebo (or placebo on sulpiride days), or 250 min after sulpiride, participants performed the RLWM task.

### Reinforcement learning working memory task

The RLWM task was presented using Psychtoolbox-3 for MATLAB. As described in the main text, participants completed two task phases: a training phase and a test phase.

In the training phase, participants were presented with stimuli in blocks of varying set sizes (between 2 and 5 stimuli in each block). Stimuli were presented one-at-a-time and participants responded with one of three button presses, assigned at random for each stimulus. Note that multiple stimuli may map to the same key press in each of the set size blocks. Participants were tasked with learning which of three buttons corresponded to each stimulus through trial-and-error. Stimuli were presented in pseudo-random order, for nine iterations, before switching to a new block. Participants completed between 2 and 3 blocks of each set size, also presented in pseudo-random order, and new stimuli were used for each block.

If participants responded correctly on a given trial, they were always given reward feedback (+1 or +2 points probabilistically: 20/80, respectively), and if they were incorrect, they were always given zero points. The number of points awarded on reward trials was not contingent on performance and, correspondingly, we found no differences between computational models fit when assuming 0,1, or 2 points for reward trials versus those fit assuming a simpler binary {1,0} for correct/incorrect trials. As such, we fit the learning algorithm (see section on "Computational modeling of behavior", below) assuming the {1,0} binary.

At the end of the training period, participants completed a surprise test phase in which pairs of stimuli were drawn from across all blocks and participants were tasked with selecting which of each pair was rewarded at a higher rate. Participants were not given feedback about the accuracy of their response in the test phase. Because participants were assigned the task of selecting which stimuli received more points during the test phase, our analyses of the test phase assumed that participants responded based on encoding reward outcomes as either 0,1, or 2 points.

One participant's data was excluded from analysis because their average late-block accuracy in the training phase was below 53% in all three sessions. A single session from another participant was excluded because they did not complete the training phase. In addition, three participants did not participate in the methylphenidate session, and one participant did not complete their placebo session. In total, 95 out of 100 methylphenidate, 97 out of 100 placebo, and 99 out of 100 sulpiride sessions were included in the final analysis of the training phase. In the test phase, an error with response logging meant that some sessions were excluded based on the criteria that we failed to capture participants' responses on at least 80% of trials. Additionally, two participants were excluded based on their choice patterns, which indicated that they merely alternated left/right presses on more than 75% of trials. In total, 69 out of 100 methylphenidate, 78 out of 100 placebo, and 80 out of 100 sulpiride sessions were included in the final analysis of the test phase.

For analysis of training phase accuracy, we included trials starting after the first correct iteration of each stimulus to avoid analyzing performance variance due to luck at the beginning of each block. Finally, for the test phase, we excluded trials with a response time of faster than 250 ms to avoid analyzing trials on which participants were merely guessing. The average of participants' median response times was 1070 ms with a standard deviation of 443 ms.

Trial-wise response accuracy in both phases was modeled with fully random, Bayesian mixed effects logistic regression models using Stan for full Bayesian inference, unless otherwise indicated. The brms package version 2.8.0 was used to fit mixed effects regression models along with R version 3.4.3. p-values were calculated from the effect estimate and standard errors estimated from the upper and lower 95% confidence intervals.

Our analysis involved two Bayesian regression models—one modeling trial-wise response accuracy on the training phase, and another modeling trial-wise accuracy during the test phase (selecting the option that received rewards at a higher rate). The first model is intended to evaluate the factors that index WM (e.g., smaller set size) versus RL contributions to learning (more previous correct iterations of a given stimulus), and how these indices are modulated by dopaminergic factors. The second model is intended to reveal the factors that make the rewards that were associated with a stimulus seem more (higher actual reward rate) versus less valuable (whether stimuli were encountered in a higher demand block).

The training phase model included independent variables which were previously shown[39] to influence choice accuracy. These included the integer set size for the block in which the stimulus was learned ($n_s$: delay), or integer number of trials since a correct response was given for each stimulus ($n_d$), the integer number of previous correct responses for that stimulus (pCor). Higher order interactions were also included to test hypotheses about how, for example, the effects of delay and set size should diminish with an increasing number of prior correct iterations.

Similarly, the test phase model included independent variables which were previously shown[39] to influence correct selection of the stimulus which was rewarded most in each pair. These included the actual difference in rewards earned for each stimulus ($\Delta V$: value difference), the mean set size of the blocks from which the two stimuli were learned ($\bar{n}_s$), and the difference in the set size from which the two stimuli were learned ($\Delta n_s$). Note that we also included a variable for the mean value of reward earned across the two stimuli in each pair ($\bar{V}$), based on work showing that higher overall value of a choice set can enhance value-based discrimination[68].

Both the training and test phase models included dopamine synthesis capacity and drug as predictors and interactions with the task variables listed here. A full list of all variables included in the model is listed in the Supplementary Tables along with their effect estimates.

### Computational modeling of behavior

We hypothesized that higher dopamine signaling may influence reliance on WM and RL processes. To test these hypotheses more precisely, we adapted a learning algorithm[10,12] involving both WM and RL modules to support stimulus-response learning and fit it to behavior which has been successfully applied to capture behavior and neural dynamics during the RLWM task in both healthy and disordered populations[8,39].

Following a previous approach[12], WM contributions were not dynamically adjusted in the algorithm, based on the inferred reliability of WM versus RL. While this elegant approach captures dynamic adjustments to WM contributions across a block, we found that doing so reduced model recoverability. Instead, we allowed WM reliance to scale linearly with the number of unique, intervening stimuli since the last correct response for each stimulus, reasoning that WM was likely to contribute less to the extent that people encounter new, competing information since their last experience with a given stimulus. Allowing WM to contribute dynamically as a function of unique, intervening items helped account for delay effects within blocks and influenced overall reliance to WM for larger versus smaller blocks.

In the model, RL proceeds by tracking action values via Rescorla-Wagner style updating. Specifically, the value of a particular action $a$ for a given stimulus $s$ (Q($s$, $a$): policy) in the RL system is updated, during learning, according to correct versus incorrect outcomes ($r$), and a learning rate parameter ($\alpha_{RL}$) that, following[8] is discounted (by $\gamma$) on incorrect trials to capture the tendency to neglect feedback about incorrect responses.

$$Q(s,a)_t \leftarrow \begin{cases} Q(s,a)_{t-1} + \alpha_{RL} \times (r_{t-1} - Q(s,a)_{t-1}) \\ Q(s,a)_{t-1} + \alpha_{RL} \times \gamma \times (r_{t-1} - Q(s,a)_{t-1}) \end{cases} \begin{matrix} if & l_{t-1}=1 \\ & r_{t-1}=0 \end{matrix} \quad (1)$$

The WM system also tracks action policies (WM($s$, $a$)), with an instantaneous effective learning rate which also discounts incorrect feedback to the same degree ($\gamma$) as the RL system.

$$WM(s,a)_t \leftarrow \begin{cases} r_{t-1} \\ WM(s,a)_{t-1} + \gamma \times (-WM(s,a)_{t-1}) \end{cases} \begin{matrix} if & r_{t-1}=1 \\ & r_{t-1}=0 \end{matrix} \quad (2)$$

We assume that participants start each block with the belief that all actions are equally likely to be correct for a given stimulus and so assign an initial value of $Q(s,a) = 1/n_a$, where $n_a$ is the number of possible actions (3) in this task. The WM system is subject to decay (rate: $\phi$) for all stimulus-action pairs that are not observed on a given trial which then decay back towards the initial value prior.

$$WM(s,a)_t \leftarrow WM(s,a)_{t-1} + \phi \times \left( \frac{1}{n_a} - WM(s,a)_{t-1} \right) \quad (3)$$

Action values from both the RL and WM systems are converted into action probabilities via the softmax function, and the respective probabilities $p_{RL}$ and $p_{WM}$ are combined linearly. Note that because the inverse temperature parameter can trade off with key parameters including the learning rate and WM reliance, we chose to fix the inverse temperature at a high value ($\beta = 50$) across all participants.

$$p = (1-\omega) \times p_{RL} + \omega \times p_{WM} \quad (4)$$

The degree to which participants rely on WM versus RL to select actions ($\omega$) is defined by the parameter $\rho$, which can be thought of as a participant's baseline propensity to rely on WM versus RL, across contexts. Overall reliance also depends on the ratio of WM capacity ($WM_{cap}$) to the number of unique, intervening stimuli encountered since the last correct response for any given stimulus, $k$, ($n_{delay,k}$). This modulation of $\rho$ allows reliance on WM to decrease dynamically, per item, as a function of delay.

$$\omega = \rho \times \min\left(1, \frac{WM_{cap}}{n_{delay,k}}\right) \quad (5)$$

Finally, to further constrain our parameter estimates, we also leverage information about choices made during the test phase. Specifically, we included a critic which learns at the same rate as the RL actor ($\alpha_{RL}$), accumulating information about the value of a stimulus (averaged across all actions) $V(s)_{RL}$.

$$V(s)_{RL,t} \leftarrow \begin{cases} V(s)_{RL,t-1} + \alpha_{RL} \times (r_{t-1} - V(s)_{RL,t-1}) \\ V(s)_{RL,t-1} + \alpha_{RL} \times \gamma \times (r_{t-1} - V(s)_{RL,t-1}) \end{cases} \begin{matrix} if & r_{t-1}=1 \ correct \\ & r_{t-1}=0 \ incorrect \end{matrix}$$
$$(6)$$

We assume that the probability that a person selects a stimulus as being rewarded the most in each pair $p$ also depends on a softmax transformation (again, with fixed $\beta = 50$) of stimulus values to $p_{V,RL}$ and

a noise term $\nu$ to accommodate guessing during the test phase.

$$p = \frac{1}{2}\nu + (1-\nu) \times p_{V,RL} \quad (7)$$

The final model thus contains six free parameters {$\alpha_{RL}$, $\gamma$, $\phi$, $\rho$, $WM_{cap}$, $\nu$}. All varied continuously except for $WM_{cap}$, which was an integer. For fitting, parameters were allowed to vary, respectively, from a minimum of {0, 0, 0, 0, 2, 0} to a maximum of {1,1,1,1,5,1}. To estimate the $WM_{cap}$, we iterated over all integers from 2 to 5, optimizing the likelihood function for all other parameters, for each iteration. We selected as each individual's WM capacity the values which produced the highest likelihood values across all integer capacity values. Parameters were estimated for each subject by initializing the action values uniformly across actions for both the WM and the RL modules.

We fit variants of the learning algorithms presented in refs. 10,12 striking a balance between the sensitivity of algorithms which allow WM contributions to vary dynamically as a function of reliability and the simplicity of algorithms which maximize parameter recoverability. We settled on a new variant (described above) in which WM reliance is a function of a baseline reliance parameter $\rho$, and the delay since the last correct iteration of a stimulus (Supplementary Methods). All variants were fit using the mfit toolbox (https://github.com/sjgershm/mfit) in MATLAB and uniform priors for the WM capacity, learning rate, and WM reliance parameters. To avoid parameters going to bounds for the WM decay, punishment neglect, and testing noise parameters, we used very slightly informative priors (beta distributions with parameters $\alpha = 1.05$ and $\beta = 1.05$). The model with recoverable parameters and the best BIC score was selected. Fit quality for the winning model was confirmed with posterior predictive checks comparing real data to data simulated from the model (Figs. S1A–C). The interpretability of our maximum a priori parameter estimates was confirmed via parameter recovery exercises where combinations of parameters values were selected at random from across the range of fitted values, data were simulated from each model, and models re-fit. Finally, re-fitted parameter values were compared to the original parameter values (Fig. S2).

The fitted model captures qualitative effects of set size and iteration number (Supplementary Fig. S1A). It also captures delay, early versus late trials and the interactions of these variables with dopamine variables including individual differences in dopamine synthesis capacity, and drugs (Supplementary Fig. S1B, C). The model not only provides an excellent fit to the data but is also recoverable, indicating that individual participant-level parameter estimates are reliable and interpretable (Supplementary Fig. S2). Prior to evaluating how parameters are altered by dopamine synthesis capacity and drugs, we removed participants with outlier values (more than ± 2 standard deviations from the mean) for parameters of interest to reduce the likelihood that any single extreme and unlikely estimate exerts too much leverage on the evaluation process. This resulted in the removal of 5 participants from the methylphenidate session (5.3% of participants), 4 from the placebo session (4.2%), and 2 from the sulpiride session (2.0%). After their removal, fitted values for the six free parameters {$\alpha_{RL}$, $\gamma$, $\phi$, $\rho$, $WM_{cap}$, $\nu$} ranged from a minimum of {.00, .039, .00, .62, 2, 0.026} to a maximum of {0.04, 0.84, 0.30, 1.00, 5, 0.14}. See the Supplementary Methods (Supplementary Fig. S3) for histograms of fitted parameter values and the cutoff for outlier values.

## PET scanning
To measure dopamine synthesis capacity, participants completed a PET scanning session using a Siemens mCT PET-CT scanner with 40 slices, 4 × 4 mm in-plane voxels, and 5 mm thick slices. Prior to scanning, participants received 185MBq (5 mCi) F-DOPA injections into an antecubital vein. To increase [$^{18}$F]-FDOPA concentrations,

participants took 150 mg carbidopa to decrease peripheral decarbox-ylase activity, and 400 mg entacapone to decrease peripheral COMT activity 1 h prior to injection. The 89-min PET scan comprised four 1-min frames, then three 2-min frames, three 3-min frames, and finally 14 5-min frames.

PET images were reconstructed with weighted attenuation cor-rection, time-of-flight correction, correction for scatter, and then smoothed with a 3 mm full-width-half-max kernel. To correct for head movement, frames were realigned to the middle frame, using SPM12. Next, images were co-registered with a structural T1-weighted MRI scan (collected in the first screening session). Presynaptic dopamine synth-esis capacity was calculated as the F-DOPA influx rate (Ki; min$^{-1}$) per voxel using the Gjedde-Patlak linear graphical analysis method for frames between 24 and 89 min and were referenced to signal in the cerebellum gray matter. FreeSurfer was used to segment each partici-pant's high-resolution anatomical MRI scans. Ki maps were normalized to MNI space and smoothed with an 8 mm full-width half-max Gaussian kernel. Finally, mean Ki values were extracted from sub-regions of the striatum, including the dorsal caudate nucleus, defined in a prior study on the basis of cortical functional connectivity patterns[69].

### Statistics and reproducibility

Our sample size was determined for the overarching project, across multiple tasks, based on the effect size of a previous pharmacological-behavioral study recently performed by our group (CMO Arnhem-Nijmegen protocol 2013/568) as described in ref. 64. In that study, 95 participants received placebo and methylphenidate on two sessions and performed a series of cognitive tasks. The effect size of that study was $r = 0.30$ ($p < 0.001$, rank correction). Using a multiple linear regression model with 7 predictor variables (7 task outcomes in over-arching project) an effect-size of $f^2 = 0.1$ (multiple regression equivalent of $r = 0.3$) we estimated that we would have 85% power (at $p = 0.05$) from a sample size of 92 subjects. We rounded our sample size up to 100 to account for potential drop-outs and technical problems.

Although, as noted, we had some data loss due to dropout and data collection errors, no data were excluded from our analyses with the exception of three participants for whom no test phase data were collected across their three drug sessions (some training phase data were collected, but these were excluded). All experiments were double-blinded and, aside from a protocol deviation described in the methods, participants completed all drug sessions in a randomized order, using a crossover design.

### Reporting summary

Further information on research design is available in the Nature Portfolio Reporting Summary linked to this article.

## Data availability

The processed data analyzed during the current study are available in the Radboud Data Repository, https://doi.org/10.34973/0apx-ck49.

## Code availability

The code for analyses and additional measures are available at: https://doi.org/10.34973/0apx-ck49.

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

## Acknowledgements

We thank our participant volunteers. This work was supported by funding from the National Institute of Mental Health (NIMH) grant R00MH125021 to A.W., National Science Foundation (NSF) grant 2336466 to A.G.E.C., and NIMH grants P50MH119467, P50MH106435, and R01 MH084840 to M.J.F., and the Dutch Research Council (NWO) grant #453-14-015 and the European Research Council (ERC) CHEM-CONTROL grant #101054532 to R.C.

## Author contributions

A.W.: conceptualization, methodology, formal analysis, writing—original draft. R.v.d.B.: investigation, formal analysis, visualization, writing—review

and editing. L.H.: investigation, writing—review and editing. D.P.: investigation, software, writing—review and editing. J.I.M.: methodology, investigation, data curation, writing—review and editing. A.G.E.C.: software, conceptualization, writing—review and editing. M.J.F.: conceptualization, supervision, formal analysis, writing—reviewing and editing. R.C.: conceptualization, supervision, formal analysis, writing—reviewing and editing.

## Competing interests

R.C. declares the existence of a competing interest, as a consultant to Roche Ltd, but holds no shares in the company. The remaining authors declare no competing interests.
