## [Peer Review file · Nature Communications]

Striatal Dopamine Can Enhance Both Fast Working Memory, and Slow Reinforcement Learning, While Reducing Implicit Effort Cost Sensitivity

Corresponding Author: Dr Andrew Westbrook

Version 0:

Reviewer comments:

Reviewer #1

(Remarks to the Author)

The manuscript "Striatal dopamine can enhance learning, both fast and slow, and also make it cheaper" studies the role of dopamine in reward learning by dissociating the effects of reinforcement learning (RL) and working memory (WM). The authors measured dopamine synthesis capacity using PET in a big sample of healthy younger participants. Moreover, all participants underwent a drug trial in which they received placebo, methylphenidate, and sulpiride across three sessions. In all sessions, participants performed a well-established reward learning task that dissociates RL and WM while using reward to learn about the appropriate actions. Correct responses resulted in higher or lower reward and participants were asked after learning to recognize those stimuli that had been associated with high reward during learning. The authors employed computational modelling of task performance to parameterize subject's reliance on WM and RL.

The results show that dopamine synthesis capacity positively predicts performance in the task by facilitating reliance on WM. Methylphenidate also increased task performance and blunted the relationship between the reliance on WM and DA synthesis capacity, which the authors interpret as methylphenidate increasing reliance on WM on participants with low synthesis capacity. At the test done after learning, Methylphenidate attenuates the fact that participants are less likely to identify stimuli associated with high reward for stimuli from sets with bigger size, which is interpreted as methylphenidate decreasing the costs associated with performing the task under high working memory load. Finally, sulpiride impaired task performance by decreasing the reliance on WM. The effects of DA synthesis capacity or the drugs on the reliance on RL are less clear in the current manuscript, although there is some evidence that it is boosted by methylphenidate as this drug boosts performance improvement after correct responses and results in higher asymptotic performance.

Because of the high sample size and the combination of PET measures of endogenous DA synthesis capacity along with pharmacological manipulations of the system, this dataset, and the results that it provides dissociating the dopaminergic modulation of WM and RL contributions on reward learning, are highly valuable. I believe that the current presentation of the methods and results is not clear and transparent enough and can be improved. I am also not completely convinced on the interpretation that methylphenidate decrease the costs. However, I have no doubt that this manuscript, upon revision, will make a substantial contribution to our understanding of how dopamine modulates reward learning.

Major points:

- 1) The methods could be clearer and more transparent.
 - a) The description of the task and analysis in the methods should include a section with all the independent variables that are analyzed with a short description of how they are derived.
 - b) The methods should also include a clearer list of all the Bayesian regressions and their factors as well as the hypothesis that they are testing.
 - c) The computational model is now presented half before the PET analysis and half after it, which unnecessarily complicates the reading. The free parameters of the models should be clearly identified and their constraints specified. Are the initial values of the WM module initialized as the ones of the RL module? Unclear how WMcap is fitted. Is gamma common across WM and RL modules? Some participants were excluded because of outlier parameter values. It would be good to know which values these participants had and how these exclusions may affect statistics. Related to outlier parameter values, why did the authors go for maximum likelihood instead of MAP or a hierarchical approach? The ranges of the fitted parameters

should also be presented.

2) Presentation of the results could be clearer and more transparent.

a. It is unclear which of the parameters of the computational model were analyzed in regression models including dopamine factors. This is important in terms of correction for multiple comparisons. At least rho and RL learning rate were analyzed. The full results of those regressions should be presented (in the main text or the supplement) instead of just some selected effects as in the current text.

b. Likewise, the full results of the regressions on accuracy restricted to early trials or late trials should be presented. Ideally, the early vs late should be included as a factor and the separation made upon constatation of a significant interaction.

3) The analysis of the test data shows that participants are less likely to identify stimuli associated with high reward when high value stimuli from sets with bigger size are compared to low value stimuli from smaller size sets, and that this effect is attenuated in the Methylphenidate condition. This finding is interpreted as methylphenidate decreasing the costs associated with performing the task under high working memory load. Again, this analysis could be presented more clearly, and I am aware that I may not have understood it fully. It strikes me, however, that the pattern of results could be related to Methylphenidate increasing attention and therefore facilitating the learning of the reward magnitude better as set sizes increases. I am not convinced that this interpretation can be ruled out in the lack of a direct measure of costs in the current task.

4) It is not very clear what is the evidence supporting that Methylphenidate increased WM reliance on participants with lower synthesis capacity. Unless I missed something in the current text, the presentation of this interpretation should be more nuanced in the abstract, the introduction, and the discussion.

5) The text would be easier to understand if some key concepts were spelled out as they are presented in the results sections. For example, how is delay operationalized? One needs to look at the methods to understand that it may refer to the number of unique intervening stimuli since the last correct response for each stimulus. Likewise, previous correct iterations would need a clear definition.

Reviewer #2

(Remarks to the Author)

The study by Westbrook et al. sought to distinguish the effects of dopamine on reinforcement learning versus dopamine on working memory. To do this they conducted a behavioral study using pharmacological and PET neuroimaging approaches in a large number (N=100) of otherwise healthy participants. Participants completed a discrimination task (stimulus-response mappings) of varying set sizes (N=2:5) that the authors argue would promote or discourage the use of working memory depending on the set size. The authors report that high dopamine synthesis and methylphenidate are associated with better performance in the task (particularly during early learning) and that low dopamine synthesis and sulpiride are associated with poorer performance in the task (although poor in this case is >80% accuracy even in sets of 5 stimuli). They then use a computational approach to understand why these changes in accuracy are occurring and report that: 1) higher dopamine synthesis is associated with greater use of WM, 2) pharmacological increases of DA with MPH is associated with a higher learning rate as evidenced by a higher incremental improvement with each rewarded trial, and 3) Blockade of D2 receptors with sulpiride reduces the working parameter by making working unreliable. The paper is well written, the experiments well designed and conducted, and the sample size is impressive for a human neuroimaging study. I thoroughly enjoyed reading the manuscript and only have a few comments for the authors outlined below:

1. Stimulus-action mappings for smaller set sizes (less than 4) are discrete and 1:1. For set sizes 4 and 5, however, stimulus-action mappings are overlapping (action 2 might be correct action for two stimuli). This seems to be introducing something other than an increasing and linear WM load as argued in the paper. Could the authors provide some insight into why they used this approach rather than independent mappings?

2. Moreover, why did they cap it at 5 stimuli? Accuracy on the task is extremely high (particularly at the end of each training block) so I wonder if lack of differences in some analyses was due to a ceiling effect.

3. The authors argue that steepness in the curve of accuracy across blocks argues for greater use of working memory in smaller set sizes. I would like to see a more quantitative approach to support this argument – perhaps fitting the data with their RL model without the WM calculations and seeing how performance at an individual level deviates from the model? I would expect that as the set size increase, a simple RL model would better fit the data. Or using their RL/WM model to show that working memory p decreases with increasing set size.

4. Argue that dopamine improves performance in the task by increasing likelihood of using WM (placebo data is in line with that) and increasing learning rate. But I don't understand how the placebo data supports the latter.

5. The blunting of cognitive effort by MPH reported at line 294 is very interesting and an important finding. However, I think the terminology of "blunting cognitive effort discounting" (e.g., focusing on the negative) may be doing a disservice to the result. It seems to me that MPH is equalizing reward values across varying cognitive efforts and that this is what is leading to the accuracy improvements following MPH administration.

6. Perhaps I am misunderstanding the methods, but it seems to me that accurate trials were probabilistically reinforced but inaccurate responses were deterministically reinforced (never rewarded). Does that mean that negative reinforcement was more informative on each trial than positive reinforcement? If this is correct, did the authors consider using their dual learning rate model (α_{rew} and α_{norew})? According to their own previous models and hypotheses one would expect sulpiride to specifically impact α_{norew} .

7. I had a difficult time understanding how the different results supported the conclusion the authors state in the abstract and the discussion. I suspect this could be improved by reworking the discussion to guide the readers through each result and then presenting their overarching hypothesis about the work at the end. I felt like I understood each piece of the paper but not the overall argument that DA is enhancing slow RL and fast WM by promoting plasticity and reducing effort sensitivity.

Reviewer #3

(Remarks to the Author)

Westbrook et al. used an established RL-WM task in combination with FDOPA PET imaging and dopaminergic pharmacological challenge in a within-subjects design in healthy controls.

The found that learning was enhanced in participants with higher dopamine synthesis capacity and in the methylphenidate condition, but impaired by sulpiride. This is a very elegant study with strong design, remarkable power and solid methods. I have the following comments.

- DA synthesis capacity was extracted from five striatal subregions, which displayed correlated signals. It would be informative to state which regions were extracted and provide a figure of the anatomical location. The rationale for using only the dorsal caudate nucleus should be given in some more detail and should not be based on analyses from the same FDOAP data set. Were similar effects on task behavior also observed for the other striatal subregions or only for the dorsal caudate nucleus?
- Analyses of choice data was performed with brms using R. It would be informative to also report leave-one-out cross-validation via the loo package.
- The authors conduct posterior predictive checks and parameter recoverability and meet high standards for computational modeling analyses. In the method section it is stated that a "wide range of models were fit via hierarchical maximum likelihood using the mfit toolbox". Please provide more details what kind of different models these were and report details on model selection!
- In the RL-WM task, steeper learning curves for smaller set sizes are thought to reflect greater reliance on WM rather than faster RL. This is a plausible assumption and has been reasoned based on previous studies using similar task variants. However, could the authors provide evidence for this claim within the data, e.g. correlation with neuropsychological measures of WM such as digit span?
- It would be helpful to publish the analyses scripts together with the paper including the R scripts for full transparency.

Version 1:

Reviewer comments:

Reviewer #1

(Remarks to the Author)

The authors have done a good job improving clarity and transparency of the manuscript and they have addressed most of my comments. There are, however, some remaining points:

The cost interpretation of the fact that participants under methylphenidate choose the most rewarding choice more frequently than under placebo when the set size difference is unfavorable to the most rewarding option is straight forward. However, I am still not completely convinced that this interpretation is so straight forward to the fact that participants choose the most rewarding option less under methylphenidate than placebo when it comes from smaller set sizes. Why would they discount the value of those options? As the points in figure 6B and C involve choices with different discriminability of value, it may be that participants under methylphenidate are better at learning the value of all the options, also those of the less rewarding option coming from the bigger set size. This may result in less subjective value difference and less probability of choosing the most rewarding one. In order to reject this explanation, the authors would need to show that there are no difference between methylphenidate and placebo in discriminability, that is on a plot like the one displayed in figure 6A.

In any case, I believe that the authors make too strong claims in the title, the abstract and conclusion about the effects of methylphenidate on effort costs considering that the evidence provided in the data is only indirect. At the least, the authors should be very clear to the reader that the data does not have any explicit measure of effort costs and that the evidence supporting their claim in the data is at least indirect and based on interpretation of previous studies.

On lines 187-193, it is not clear whether the results are corrected or uncorrected. The text is easily read as if they were corrected.

Finally, it is comforting to see that the removal of outliers does not fully drive the reported correlations between synthesis capacity and model parameters. However, considering that the exclusion criteria is rather harsh (2 SD instead of the more standard 3SD) and implies exclusion of a substantial number of participants (5%), I would suggest that the results including

the outliers are reported in the supplement. In this way, the reader will be aware of the impact of outlier removal. An alternative to avoid exclusion of so many participants would be to normalize the data.

(Remarks on code availability)

Reviewer #2

(Remarks to the Author)

The authors addressed all my concerns and the manuscript has improved significantly.

(Remarks on code availability)

We thank the reviewers for their work considering our prior submission. In the following, we detail a point-by-point response, indicating additional analyses and revisions to our manuscript and supplement. Reviewer Comments are displayed in **bold and highlighted** font, and our response is in regular font.

REVIEWER COMMENTS

Reviewer #1 (Remarks to the Author):

The manuscript "Striatal dopamine can enhance learning, both fast and slow, and also make it cheaper" studies the role of dopamine in reward learning by dissociating the effects of reinforcement learning (RL) and working memory (WM). The authors measured dopamine synthesis capacity using PET in a big sample of healthy younger participants. Moreover, all participants underwent a drug trial in which they received placebo, methylphenidate, and sulpiride across three sessions. In all sessions, participants performed a well-established reward learning task that dissociates RL and WM while using reward to learn about the appropriate actions. Correct responses resulted in higher or lower reward and participants were asked after learning to recognize those stimuli that had been associated with high reward during learning. The authors employed computational modelling of task performance to parameterize subject's reliance on WM and RL.

The results show that dopamine synthesis capacity positively predicts performance in the task by facilitating reliance on WM. Methylphenidate also increased task performance and blunted the relationship between the reliance on WM and DA synthesis capacity, which the authors interpret as methylphenidate increasing reliance on WM on participants with low synthesis capacity. At the test done after learning, Methylphenidate attenuates the fact that participants are less likely to identify stimuli associated with high reward for stimuli from sets with bigger size, which is interpreted as methylphenidate decreasing the costs associated with performing the task under high working memory load. Finally, sulpiride impaired task performance by decreasing the reliance on WM. The effects of DA synthesis capacity or the drugs on the reliance on RL are less clear in the current manuscript, although there is some evidence that it is boosted by methylphenidate as this drug boosts performance improvement after correct responses and results in higher asymptotic performance.

Because of the high sample size and the combination of PET measures of endogenous DA synthesis capacity along with pharmacological manipulations of the system, this dataset, and the results that it provides dissociating the dopaminergic modulation of WM and RL contributions on reward learning, are highly valuable. I believe that the current presentation of the methods and results is not clear and transparent enough and can be improved. I am also not completely convinced on the interpretation that methylphenidate decrease the costs. However, I have no doubt that this manuscript, upon revision, will make a substantial contribution to our understanding of how dopamine modulates reward learning.

We are grateful for the positive overall assessment of our dataset and our experimental design.

Major points:

1) The methods could be clearer and more transparent.

a) The description of the task and analysis in the methods should include a section with all the independent variables that are analyzed with a short description of how they are derived.

We agree that the methods could be both clearer and now transparent. We have made modifications to improve on both fronts. First, we now include a section in the methods with all the independent variables and how they are derived and their basis of selection. The new text on lines 615-620 is as follows:

“The training phase model included independent variables which were previously shown (Collins et al., 2017) to influence choice accuracy. These included the integer set size for the block in which the stimulus was learned (n_s) the “delay”, or integer number of trials since a correct response was given for each stimulus (n_d), the integer number of previous correct responses for that stimulus ($pCor$). Higher order interactions were also included to test hypotheses about how, for example, the effects of delay and set size should diminish with an increasing number of prior correct iterations.

Similarly, the test phase model included independent variables which were previously shown (Collins et al., 2017) to influence correct selection of the stimulus which was rewarded most in each pair. These included the actual difference in rewards earned for each stimulus or “value difference” (ΔV), the mean set size of the blocks from which the two stimuli were learned (\bar{n}_s), and the difference in the set size from which the two stimuli were learned (Δn_s). Note that we also included a variable for the mean value of reward earned across the two stimuli in each pair (\bar{V}), based on work showing that higher overall value of a choice set can enhance value-based discrimination (Smith and Krajbich, 2018). Additionally, this overall value term was added to the model because we were interested in how dopamine (which is released in greater quantities when the overall value of cues is greater) biases learning in the direct versus indirect pathways. We have studied this learning bias induced by dopaminergic signaling using the Probabilistic Selection Task in prior work (e.g. Frank, Seeberger, O’Reilly, 2004).

Both the training and test phase models included dopamine synthesis capacity and drug as predictors and interactions with the task variables listed here. A full list of all variables included in the model is listed in the Supplement along with their effect estimates.”

Frank, M. J., Seeberger, L. C. & O’Reilly, R. C. By Carrot or by Stick: Cognitive Reinforcement Learning in Parkinsonism. *Science* **306**, 1940–1943 (2004).

Smith, S. M. & Krajbich, I. Gaze Amplifies Value in Decision Making. *Psychological Science* **30**, 116–128 (2018).

b) The methods should also include a clearer list of all the Bayesian regressions and their factors as well as the hypothesis that they are testing.

Next, we have now also included language listing the Bayesian regressions as well as their core hypotheses, on lines 607-614:

“Our analysis involved two Bayesian regression models – one modeling trial-wise response accuracy on the training phase, and another modeling trial-wise accuracy during the test phase (selecting the option which received rewards at a higher rate). The first model is intended to

evaluate the factors which index working memory (e.g. smaller set size) versus reinforcement learning contributions to learning (more previous correct iterations of a given stimulus), and how these indices are modulated by dopaminergic factors. The second model is intended to reveal the factors which make the rewards that were associated with a stimulus seem more (higher actual reward rate) versus less valuable (whether stimuli were encountered in a higher demand block).”

c) The computational model is now presented half before the PET analysis and half after it, which unnecessarily complicates the reading.

We agree that splitting up a description of the computational modeling methods was inefficient and complicated the reading. We have now moved the PET analysis section to the end of the methods section and integrated the modeling methods description, much improving readability.

The free parameters of the models should be clearly identified and their constraints specified. Are the initial values of the WM module initialized as the ones of the RL module? Unclear how WMcap is fitted.

We regret omitting key details in our methods. We now include the following text on lines 692-699:

“The final model thus contains six free parameters $\{\alpha_{RL}, \gamma, \phi, \rho, WM_{cap}, \nu\}$. All varied continuously except for WM_{cap} , which was integer. For fitting, parameters were allowed to vary, respectively, from a minimum of $\{0, 0, 0, 0, 2, 0\}$ to a maximum of $\{1, 1, 1, 1, 5, 1\}$. To estimate the WM_{cap} , we iterated over all integers from 2 to 5, optimizing the likelihood function for all other parameters, for each iteration. We selected as each individual’s working memory capacity the values which produced the highest likelihood values across all integer capacity values. Parameters were estimated for each subject by initializing the action values uniformly across actions for both the WM and the RL modules.”

Is gamma common across WM and RL modules?

Yes, the gamma parameter was common across the WM and RL models. We reasoned that neglect of punishments versus rewards would have a similar impact on the information available to update both the RL and the WM modules. We have now made this explicit in the manuscript in lines 654-655:

“The WM system also tracks action policies ($WM(s, a)$), with an instantaneous effective learning rate which also discounts incorrect feedback to the same degree (γ) as the RL system.”

Some participants were excluded because of outlier parameter values. It would be good to know which values these participants had and how these exclusions may affect statistics. The ranges of the fitted parameters should also be presented.

The following histograms show the distributions of parameter values across individuals and drugs sessions, we also plot red lines indicating the 2 SD cutoff used for identifying outlying parameter values. We have now included these figures in the Supplement.

The removal of outliers did not affect any of the core inferences in the manuscript. There are some changes in exact p-values when we include sessions with outlier values for two parameters (WM decay ϕ , and WM capacity, WM_{cap}). However, in neither case do the inferences which we make change fundamentally.

Regarding the WM decay term (ϕ), if we do not remove sessions with outlier value estimates, the overall positive relationship between DA synthesis capacity and WM reliance (ρ) across sessions is no longer significant ($\beta = .16$; $p = .10$). However, positive correlations remain between DA synthesis capacity and WM reliance in the placebo ($r = .24$; $p = .028$) and sulpiride sessions ($r = .28$; $p = .0098$). Similarly, the relationship between dopamine synthesis capacity and the learning rate is no longer significant in the methylphenidate session ($\beta = .17$; $p = .13$). However, a two-way interaction between dopamine synthesis capacity and methylphenidate remains, indicating that the relationship between the learning rate and dopamine synthesis capacity is stronger on methylphenidate versus placebo ($\beta = .28$; $p = .032$). Thus, in both cases, including outlier WM decay parameter values does not fundamentally alter the interpretation, even if there are small changes to the resulting p-values.

When we do not remove outlying WM capacity values (i.e., $WM_{cap} = 2$), the overall positive relationship between DA synthesis capacity and WM reliance across sessions is no longer significant ($\beta = .19$; $p = .062$). However, positive correlations remain between DA synthesis capacity and WM reliance in the placebo ($r = .22$; $p = .041$) and sulpiride sessions ($r = .28$; $p = .0098$) separately, indicating that not excluding on the basis of outlying WM capacity also does not fundamentally alter these inferences.

Related to outlier parameter values, why did the authors go for maximum likelihood instead of MAP or a hierarchical approach?

There was an error in the original manuscript which we regret. MAP estimates are used, not maximum likelihood. We used the mfit toolbox from Sam Gershman which allows for setting priors and deriving MAP estimates. We did not have a strong methodological motivation for using MAP rather than fitting a fully hierarchical Bayesian model, but our parameter estimates seemed well behaved in that they were recoverable and reproduced key behavioral features, so we were satisfied by the simpler MAP approach.

2) Presentation of the results could be clearer and more transparent.

a. It is unclear which of the parameters of the computational model were analyzed in regression models including dopamine factors. This is important in terms of correction for multiple comparisons. At least ρ and RL learning rate were analyzed. The full results of those regressions should be presented (in the main text or the supplement) instead of just some selected effects as in the current text.

We agree that the full results of these regression models should be presented. Our a priori hypotheses implicated dopamine factors in individual differences in three parameters: the learning rate α_{RL} and WM reliance ρ , and working memory capacity WM_{cap} because of prior work (including our own) linking individual differences to various working memory capacity measures (e.g. Cools et al. 2008; van der Schaaf et al., 2013). We now include the full regression details for these tests in the Supplement (Supplemental Tables S3-S5). We now also correct for multiple comparisons, including corrected and uncorrected p-values and indicating which survive Bonferroni correction.

In the main text, regarding the main effect of dopamine synthesis capacity on WM reliance (ρ), we now stipulate on lines 190-194:

“We note that the overall effect of dopamine synthesis capacity across sessions does not survive correction for multiple comparisons across all model parameters regressed onto dopamine factors ($p_{\text{Bonferroni}} = .087$; Supplemental Tables S3–S5). Nevertheless, the overall effect estimate reflects separate, significant positive correlations between dopamine synthesis capacity and working memory reliance in the placebo ($r = .24$; $p = .028$) and sulpiride sessions ($r = .28$; $p = .0098$).”

Regarding the two-way interaction between the methylphenidate and dopamine synthesis capacity predicting the learning rate α_{RL} , we now stipulate on lines 219-224:

“A hierarchical regression of the learning rate parameter α_{RL} on dopamine, controlling for session number, reveals a two-way interaction such that methylphenidate boosts learning rates for those with higher dopamine synthesis capacity ($\beta = .30$; $p = .032$; albeit this effect does not survive correction for multiple comparisons $p_{\text{Bonferroni}} = .096$; Supplemental Tables S3–S5).”

Cools, R., Gibbs, S. E., Miyakawa, A., Jagust, W. & D’Esposito, M. Working Memory Capacity Predicts Dopamine Synthesis Capacity in the Human Striatum. *J Neurosci* **28**, 1208–1212 (2008).

Schaaf, M. E. van der, Fallon, S. J., Huurne, N. ter, Buitelaar, J. & Cools, R. Working Memory Capacity Predicts Effects of Methylphenidate on Reversal Learning. *Neuropsychopharmacol* **38**, 2011–2018 (2013).

b. Likewise, the full results of the regressions on accuracy restricted to early trials or late trials should be presented. Ideally, the early vs late should be included as a factor and the separation made upon constataion of a significant interaction.

We now present the full regressions on accuracy for early and late trials in Supplemental Tables S6–S8. We agree that an early versus late distinction should be made on constataion of a significant interaction, which the data do not support:

“A hierarchical logistic regression, restricted to early trials, regressing accuracy on set size, drug, and dopamine synthesis capacity reveals that participants with higher dopamine synthesis capacity perform better when WM plays a bigger role ($\beta = .14$; $p = .019$; Supplemental Table S6). Although the same effect is not significant in late trials ($\beta = .12$; $p = .18$; Supplemental Table S6), the data do not support a distinction between early and late trials. The two-way interaction between dopamine synthesis capacity and early versus late trials is not significant ($\beta = -0.0099$; $p = .89$; Supplemental Table S8).”

3) The analysis of the test data shows that participants are less likely to identify stimuli associated with high reward when high value stimuli from sets with bigger size are compared to low value stimuli from smaller size sets, and that this effect is attenuated in the Methylphenidate condition. This finding is interpreted as methylphenidate decreasing the costs associated with performing the task under high working memory load. Again, this analysis could be presented more clearly, and I am aware that I may not have understood it fully. It strikes me, however, that the pattern of results could be related to Methylphenidate increasing attention and therefore facilitating the learning of the reward magnitude better as set sizes increases. I am not convinced that this interpretation can be ruled out in the lack of a direct measure of costs in the current task.

We do not think this alternative explanation could account for the effects of methylphenidate on choice as a function of the difference in set sizes. In short, we think this account is unlikely because we find that methylphenidate both increases choice accuracy (when the more rewarded option comes from a larger size set) and decreases choice accuracy (when the more rewarded option comes from a smaller size set). It is not that methylphenidate sensitizes reward learning in general, or even that methylphenidate sensitizes reward learning as set size increases (because these effects marginalize over pair-averaged set sizes).

Nevertheless, we agree with the reviewer that our measure is not an explicit measure of effort costs, but rather an implicit measure. We note that this effect – that set size “discounts” the learned reward values (replicating Collins et al. (2017)) – is consistent with our prior work using explicit effort cost measures. For example, in Westbrook et al. (2020) we showed that larger working memory demands are explicitly subjectively costly – causing participants to discount offers to repeat tasks for money. Importantly, in that study, we also showed that striatal dopamine made people more sensitive to effort benefits and less sensitive to explicit effort costs. As such, we see the present result as a conceptual replication of that finding, albeit with an implicit cost measure.

Our interpretation is also grounded in a very similar implicit learning effect reported in Cavanagh et al. (2014) where instances of cognitive conflict, in a Simon task, made participants learn to treat rewards associated with the corresponding stimulus as less rewarding. In this prior study, the implicit cost learning effect was modulated by genetic and pharmacological dopamine factors.

Collins, A. G. E., Albrecht, M. A., Waltz, J. A., Gold, J. M. & Frank, M. J. Interactions Among Working Memory, Reinforcement Learning, and Effort in Value-Based Choice: A New Paradigm and Selective Deficits in Schizophrenia. *Biol Psychiat* **82**, 431–439 (2017).

Westbrook, A. *et al.* Dopamine promotes cognitive effort by biasing the benefits versus costs of cognitive work. *Science* **367**, 1362–1366 (2020).

4) It is not very clear what is the evidence supporting that Methylphenidate increased WM reliance on participants with lower synthesis capacity. Unless I missed something in the current text, the presentation of this interpretation should be more nuanced in the abstract, the introduction, and the discussion.

On reflection, we agree with the Reviewer that this claim was overstated. We have seen in multiple datasets, including our own prior work, the pattern by which methylphenidate has a bigger impact among those with lower dopamine synthesis capacity (indeed we also found it in the Westbrook et al. (2020) study cited above, in these same participants). However, the evidence we have here (the sign of a non-significant interaction between dopamine synthesis capacity and methylphenidate) is weak and does not warrant such a claim on its own. This claim has been removed from the discussion and conclusions section from lines 193-197:

“...significant positive correlations between dopamine synthesis capacity and working memory reliance in the placebo ($r = .24$; $p = .028$) and sulpiride sessions ($r = .28$; $p = .0098$). In contrast, there was no correlation between dopamine synthesis capacity and WM reliance on

methylphenidate ($r = .028$; $p = .80$; Figure 4A). However, the dopamine synthesis by drug interaction ($\beta = -.20$; $p = .23$) is not significant.”

Westbrook, A. *et al.* Dopamine promotes cognitive effort by biasing the benefits versus costs of cognitive work. *Science* **367**, 1362–1366 (2020).

5) The text would be easier to understand if some key concepts were spelled out as they are presented in the results sections. For example, how is delay operationalized? One needs to look at the methods to understand that it may refer to the number of unique intervening stimuli since the last correct response for each stimulus. Likewise, previous correct iterations would need a clear definition.

We thank the reviewer for an excellent suggestion. We have now included definitions on lines 88-89:

“...the number of items and delays (defined here as the number of trials since the last correct response for a given stimulus) grow to exceed WM capacity...”

Lines 100-101:

“...previous iteration count (the number of times a stimulus has been encountered)...”

And lines 156-158:

“In our model we also simultaneously estimate the effects of set size, the number of previous correct iterations (the number of prior trials on which a correct response was given for each stimulus), and the delay (number of trials) since the last correct iteration...”

Reviewer #2 (Remarks to the Author):

The study by Westbrook *et al.* sought to distinguish the effects of dopamine on reinforcement learning versus dopamine on working memory. To do this they conducted a behavioral study using pharmacological and PET neuroimaging approaches in a large number (N=100) of otherwise healthy participants. Participants completed a discrimination task (stimulus-response mappings) of varying set sizes (N=2:5) that the authors argue would promote or discourage the use of working memory depending on the set size. The authors report that high dopamine synthesis and methylphenidate are associated with better performance in the task (particularly during early learning) and that low dopamine synthesis and sulpiride are associated with poorer performance in the task (although poor in this case is >80% accuracy even in sets of 5 stimuli). They then use a computational approach to understand why these changes in accuracy are occurring and report that: 1) higher dopamine synthesis is associated with greater use of WM, 2) pharmacological increases of DA with MPH is associated with a higher learning rate as evidenced by a higher incremental improvement with each rewarded trial, and 3) Blockade of D2 receptors with sulpiride reduces the working parameter by making working unreliable. The paper is well written, the experiments well designed and conducted, and the sample size is impressive for a human neuroimaging study. I thoroughly enjoyed reading the manuscript and only have a few comments for the authors outlined below:

We are grateful for the positive overall assessment of our dataset and our manuscript.

1. Stimulus-action mappings for smaller set sizes (less than 4) are discrete and 1:1. For set sizes 4 and 5, however, stimulus-action mappings are overlapping (action 2 might be correct action for two stimuli). This seems to be introducing something other than an increasing and linear WM load as argued in the paper. Could the authors provide some insight into why they used this approach rather than independent mappings?

In fact, stimulus-action mappings were not 1:1 for smaller set sizes. The task was designed so that there could be overlapping keys even for set size 2 blocks. We regret not making this clearer in the original submission and have now edited the methods to do so on lines 564-567.

“In the training phase, participants were presented with stimuli in blocks of varying set sizes (between 2 and 5 stimuli in each block). Stimuli were presented one-at-a-time and participants responded with one of three button presses which were assigned at random for each stimulus. Note that multiple stimuli may map to the same key press in each of the set size blocks.”

2. Moreover, why did they cap it at 5 stimuli? Accuracy on the task is extremely high (particularly at the end of each training block) so I wonder if lack of differences in some analyses was due to a ceiling effect.

The Reviewer is correct that, in prior studies, we used a larger range of set sizes (up to $n_s = 6$). In this study, we capped set sizes at $n_s = 5$ for practical reasons. These data were collected as part of a larger pharmaco-imaging study and we were constrained by the amount of time that could be devoted to this task while participants were still under the influence of the drugs. As such, prior to collecting data, we optimized task design through simulation studies to ensure that we could capture key effects and found that they should be detectable using only set sizes of $n_s = 2$ through $n_s = 5$. That said, we agree that this cap may have limited our ability to detect other kinds of effects – e.g. with respect to dopamine factors.

It is conceivable that we might have had greater range for detecting certain effects if we had used a wider range of set sizes. If we had used set size 6, for example, we anticipate that RL would have played a larger role in for high set size blocks. Greater reliance on RL, in turn, might have helped use detect an effect of dopamine synthesis capacity on the RL learning rate, and not just the contrast of methylphenidate and placebo. We now note this possibility in the discussion:

“It is also conceivable that dopamine synthesis capacity might boost RL learning rates as a main effect, but our study design was insensitive to this relationship. Future work could explore a wider range of set sizes (e.g. set size 6) where performance would depend to an even greater extent on RL mechanisms.”

3. The authors argue that steepness in the curve of accuracy across blocks argues for greater use of working memory in smaller set sizes. I would like to see a more quantitative approach to support this argument – perhaps fitting the data with their RL model without the WM calculations and seeing how performance at an individual level deviates from the model? I would expect that as the set size increase, a simple RL model would better fit the data. Or using their RL/WM model to show that working memory p decreases with increasing set size.

We certainly agree with the logic of this point. In fact, this basic approach was taken in prior work with the RLWM task. For example, Collins et al. (2012) showed that a simple RL model could be fit to any individual set size. The key result, however, is that a single RL model could not be used to fit ALL set sizes, unless learning rates were allowed to vary for each set size (which is not parsimonious and still doesn't capture all the trial-to-trial effects e.g. of delay and set size interactions with each other and with previous correct). Inferring that WM plays a growing role for smaller set sizes was taken as a more parsimonious account of learning dynamics than assuming that the RL learning rate changes as a function of set size. Several other studies since have also showed that the data are best fit by the hybrid WMRL model and not by either an RL only or WM only module, even when each of these are allowed to have additional free parameters (e.g. Collins et al 2014). Thus, by this very logic, prior work has supported increasing WM involvement for smaller set sizes.

We also note evidence from our logistic regression analyses which supports decreasing relative involvement of WM with increasing set size. For example, we find that the adverse effect of delay on trial-level accuracy grows with increasing set size – consistent with a capacity-limited WM system (i.e., interference between items within limited capacity can explain this pattern). Importantly, we also find that a marker of RL dynamics – the number of previous correct iterations on trial-level accuracy – grows with increasing set sizes. That is, the number of previous correct iterations predicts future accuracy, to a greater degree, in larger set sizes, despite lower trial-level accuracy for larger set size blocks. We infer that previous correct iterations predict future accuracy more because RL plays an increasing role as set size increases, trading off against a decreasing role of WM.

Finally, such behavioral evidence is also buttressed by neural data. Collins & Frank (2018) decoded trial-level EEG data across set sizes and found separable spatiotemporal patterns of signal reflecting WM involvement (incorporating set size and delay effects) and reflecting RL involvement (incorporating a WM-free Q value from a simple RL model) during each trial. Moreover, these dynamics jointly predicted trial-level PEs at the time of feedback and traded off such that when WM played a larger role, RL-based feedback PE's were smaller. This tradeoff further shifted with set size such that in larger set sizes, feedback PE's were larger, leading to greater accumulation of neural signatures of Q values. These effects were replicated in Rac-Lubashevsky et al. 2023, who further showed that these neural signatures of enhanced RL under high WM load predicted *better* subsequent retention of learned stimulus-response mappings for those stimuli experienced under load. The authors interpreted these data as showing that WM and RL not only trade off against each other, but that they trade off because of a cooperative interaction: for small set sizes when WM can maintain S-R mappings, the system makes better predictions which cooperatively inform the RL system. Better predictions, in turn, result in smaller prediction errors and thus paradoxically slower RL in smaller set size blocks. In contrast, in larger set size blocks, the WM system cannot contribute as much, and the system relies more on RL to guide behavior across trials. Because the RL system is not subject to active WM memory it can support better retention in the later test phase.

Collins, A. G. E. & Frank, M. J. How much of reinforcement learning is working memory, not reinforcement learning? A behavioral, computational, and neurogenetic analysis. *European Journal of Neuroscience* **35**, 1024–1035 (2012).

Collins, A. G. E., Brown, J. K., Gold, J. M., Waltz, J. A. & Frank, M. J. Working Memory Contributions to Reinforcement Learning Impairments in Schizophrenia. *J Neurosci* **34**, 13747–13756 (2014).

Collins, A. G. E. & Frank, M. J. Within- and across-trial dynamics of human EEG reveal cooperative interplay between reinforcement learning and working memory. *Proceedings of the National Academy of Sciences* **115**, 2502–2507 (2018).

Rac-Lubashevsky, R., Cremer, A., Collins, A.G.E., Frank, M.J.*. & Schwabe, L.*. Neural index of reinforcement learning predicts improved stimulus-response retention under high working memory load. *Journal of Neuroscience* **43**, 3131–3143 (2023).

4. Argue that dopamine improves performance in the task by increasing likelihood of using WM (placebo data is in line with that) and increasing learning rate. But I don't understand how the placebo data supports the latter.

We agree with the Reviewer's perspective on this point. Indeed, if dopamine increases the learning rate, in general, we would predict that higher dopamine synthesis capacity (on placebo) also boosts learning rates, like methylphenidate did. However, we did not find this; instead we found that dopamine synthesis capacity is unrelated to the learning rate parameter.

We do not know why dopamine synthesis capacity did not boost the learning rate in our dataset. One potential explanation relates to whether dopamine dynamics do or do not match the time scales that are relevant for learning versus implicit decision-making about WM. Perhaps individual differences in dopamine synthesis capacity, on its own, are modulate tonic dopamine levels but are insufficient to drive individual differences in phasic dopamine signaling needed for incremental RL. Thus, even if differences in dopamine synthesis capacity correlate with individual differences in tonic DA signaling, they do not produce phasic differences and may be therefore incapable of driving individual differences in phasic prediction-error based learning signals.

Another possibility has to do with the cooperative interaction between the WM and the RL systems, which might bias estimates of the learning rate as a function of WM reliance. Namely, if dopamine synthesis capacity boosts WM reliance, then we would predict smaller prediction errors driving the RL system for people with higher dopamine synthesis capacity. Smaller prediction errors result from a cooperative interaction between the WM and RL systems (previously shown in both behavioral and neural data by Collins et al., 2018 and Rac-Lubashevsky et al. 2023) where RL is slowed because the WM system makes better predictions, especially when managing smaller set sizes. As such, the learning rate values we estimate might be biased lower for individuals who rely on WM more, and this might mask any countermanding effect of dopamine synthesis capacity on higher learning rates.

We also speculate that dopamine synthesis capacity does not predict individual differences in learning rates because of systemic adaptation of dopaminergic signaling systems. It is conceivable that the brains of individuals who synthesize dopamine at a higher rate also down-regulate dopamine receptor density in the course of development to “balance out” stronger signaling capacity, thereby moderating learning rates. As such, an acute perturbation like methylphenidate may boost learning rates, but “chronically” elevated dopamine synthesis capacity may not.

Finally, a more pedestrian explanation is that we simply did not have the power to detect an individual difference effect on learning rates. As our analysis reveals, effective learning rates are quite small once the contributions of a WM system are taken into account. It is possible that much larger sample sizes are needed to detect an individual difference effect on small learning rates than we were powered to detect.

Collins, A. G. E. & Frank, M. J. Within- and across-trial dynamics of human EEG reveal cooperative interplay between reinforcement learning and working memory. *Proceedings of the National Academy of Sciences* **115**, 2502–2507 (2018).

Rac-Lubashevsky, R., Cremer, A., Collins, A.G.E., Frank, M.J.*. & Schwabe, L.*. Neural index of reinforcement learning predicts improved stimulus-response retention under high working memory load. *Journal of Neuroscience* **43**, 3131–3143 (2023).

5. The blunting of cognitive effort by MPH reported at line 294 is very interesting and an important finding. However, I think the terminology of “blunting cognitive effort discounting” (e.g., focusing on the negative) may be doing a disservice to the result. It seems to me that MPH is equalizing reward values across varying cognitive efforts and that this is what is leading to the accuracy improvements following MPH administration.

We appreciate the Reviewer’s inclination to focus on positive over negative effects. However, we note that there is an intriguing *positive* effect of demand level on reward learning (Collins et al., 2017) which we wish to highlight in this study. One of the primary reasons we were interested in this task is that we wanted to study the effects of effort on reward learning (and modulatory effects of dopamine) just as we have done in the decision-making domain (Westbrook et al., 2019; Westbrook et al., 2020). The positive effect, in the RLWM task, is that increasing set size makes rewards learned in the context of higher demands seem less rewarding. We acknowledge, as noted by R1, that our index of effort costs is merely implicit, unlike our prior study where we ask participants to make explicit cost-benefit decisions between high and low working memory load levels. Nevertheless, our implicit and explicit measures converge on the conclusion that higher memory demands are subjectively costly and this can explicitly and implicitly discount the value of rewards during learning and decision-making.

We note that our implicit discounting measure reflects a positive, adverse effect of load levels on perceived reward value in both the placebo and methylphenidate sessions. Consistent with the Reviewer’s perspective, the moderating effect of methylphenidate does indicate a kind of increase in accuracy – namely, our participants’ assessment of relative reward rates is less biased by the irrelevant dimension of set size. However, this pattern does not reflect an overall increase in the accuracy of reward value. Indeed, considering only choices in which the options which were rewarded more highly came from smaller set size blocks, the probability of choosing correctly decreases, on average (left half of Figure 6B). Thus, methylphenidate both decreases and increases choice accuracy, depending on the relative difference in set size.

Collins, A. G. E., Albrecht, M. A., Waltz, J. A., Gold, J. M. & Frank, M. J. Interactions Among Working Memory, Reinforcement Learning, and Effort in Value-Based Choice: A New Paradigm and Selective Deficits in Schizophrenia. *Biol Psychiat* **82**, 431–439 (2017).

Westbrook, A., Lamichhane, B. & Braver, T. The Subjective Value of Cognitive Effort is Encoded by a Domain-General Valuation Network. *J Neurosci* **39**, 3934–3947 (2019).

Westbrook, A. *et al.* Dopamine promotes cognitive effort by biasing the benefits versus costs of cognitive work. *Science* **367**, 1362–1366 (2020).

6. Perhaps I am misunderstanding the methods, but it seems to me that accurate trials were probabilistically reinforced but inaccurate responses were deterministically reinforced (never rewarded). Does that mean that negative reinforcement was more informative on each trial than positive reinforcement? If this is correct, did the authors consider using their dual learning rate model (alpha_rew and alpha_norew)? According to their own previous models and hypotheses one would expect sulpiride to specifically impact alpha_norew.

We apologize for the confusion on this point. In fact, both correct and incorrect trials are equally informative because outcomes are deterministic – if participants were correct, they always received a reward, and if they were incorrect, they always received a punishment. We have now amended the methods to make this clear on lines 572-573:

“If participants responded correctly on a given trial, they were always given reward feedback (+1 or +2 points probabilistically: 20/80, respectively), and if they were incorrect, they were always given zero points.”

We have also altered the caption for Figure 1 to state:

“If they respond correctly, they are rewarded points (+2 or +1, amounts determined probabilistically, see Methods for full details) and if they are incorrect, they receive no points.”

Note that although the *amount* of reward that participants earn for being correct is probabilistic, the reward magnitude distinction (+1 versus +2) does not matter for determining the optimal policy.

Instead of using a dual learning rate model, we did explicitly model a neglect of punishments versus rewards, following prior work (e.g. Master *et al.* 2020) in which a neglect term was used because it made the model more identifiable than having two learning rates. Prompted by the Reviewers comment, we performed a t-test of the contrast of neglect parameters in the placebo and sulpiride session, but do not detect a significant difference ($p = 0.63$).

Master, S. L. *et al.* Disentangling the systems contributing to changes in learning during adolescence. *Dev Cogn Neuros* **41**, 100732 (2020).

7. I had a difficult time understanding how the different results supported the conclusion the authors state in the abstract and the discussion. I suspect this could be improved by reworking the discussion to guide the readers through each result and then presenting their overarching hypothesis about the work at the end. I felt like I understood each piece of the paper but not the overall argument that DA is enhancing slow RL and fast WM by promoting plasticity and reducing effort sensitivity.

We appreciate the observation that our core message did not come through clearly. We have reworked key parts of the discussion and conclusion. Specifically, in the Discussion, we now state on lines 375-377:

“We find that striatal dopamine signaling – reflecting either dopamine synthesis capacity or dopamine drugs – can promote learning, with distinct effects on both RL and WM...”

And on lines 382-386:

“Importantly, we find that dopamine promotes both WM and RL processes when controlling for the effects of dopamine on the other. Thus, we infer that striatal dopamine signaling enhances learning, both fast (via increased reliance on fast and flexible WM) and slow (by boosting relatively slow, but practically unlimited capacity RL).”

And on lines 411-414:

“Taken together with our prior study, our present results support the hypotheses that striatal dopamine signaling can make WM tasks less effort costly, both as people learn about the costs of performing the task, and at the moment of choice, when they decide whether to expend effort in the future.”

Finally, in the Conclusion, we now write on lines 497-505:

“These effects indicate that striatal dopamine can increase reliance on “fast” and flexible working memory for learning. Yet, even after accounting for the effects of dopamine on working memory, we find evidence that striatal dopamine can accelerate “slow” learning processes as well. Namely, we find that methylphenidate accelerates reinforcement learning rates, consistent with the hypothesis that striatal dopamine promotes plasticity on slower time scales too. Finally, we find evidence that pharmacological enhancement of striatal dopamine signaling can blunt effort cost learning that happens when people perform demanding tasks. This result complements prior work by showing that dopamine not only influences effort-based decision-making at the time of choice, but also by shaping the how people learn about effort costs in the first place.”

Reviewer #3 (Remarks to the Author):

Westbrook et al. used an established RL-WM task in combination with FDOPA PET imaging and dopaminergic pharmacological challenge in a within-subjects design in healthy controls. They found that learning was enhanced in participants with higher dopamine synthesis capacity and in the methylphenidate condition, but impaired by sulpiride. This is a very elegant study with strong design, remarkable power and solid methods. I have the following comments.

We are grateful for the positive overall assessment of our experimental design and our analyses.

• DA synthesis capacity was extracted from five striatal subregions, which displayed correlated signals. It would be informative to state which regions were extracted and provide a figure of the anatomical location. The rationale for using only the dorsal caudate nucleus should be given in some more detail and should not be based on analyses from the same FDOAP data set. Were similar effects on task behavior also observed for the other striatal subregions or only for the dorsal caudate nucleus?

The dorsal caudate nucleus was chosen based on prior work linking this region to higher order cognitive regions and cognitive function (e.g. including the dorsolateral prefrontal cortex; Haber et al., 2009) and also work implicating the dorsal caudate nucleus, in particular, in higher-order RL processes (e.g. Badre & Frank, 2012) and working memory gating (e.g. Fallon et al., 2017).

The specific boundaries of the five striatal sub-regions were not derived from the current FDOPA dataset but instead came from a parcellation of the human striatum based on the clustering of functional connectivity patterns (Piray et al. 2017). This analysis yielded five clusters: the dorsal caudate nucleus, the medial caudate nucleus, the ventral striatum, the posterior putamen, and the anterior putamen. We have now included a description of the dorsal caudate nucleus used in the present study, in the Supplement.

“Description of the dorsal caudate nucleus used for PET analyses

For our analyses of the effect of individual differences in dopamine synthesis capacity, we used the dorsal caudate nucleus (green region in Figure S5) as independently defined from a study which partitioned the striatum based on cortical connectivity patterns (Piray et al., 2017). As noted, the dorsal caudate nucleus was chosen based on prior work linking this region to higher order cognitive regions and cognitive function (e.g. including the dorsolateral prefrontal cortex; Haber et al., 2009) and work implicating the dorsal caudate nucleus, in particular, in higher-order RL processes (e.g. Badre & Frank, 2012) and working memory gating (e.g. Fallon et al., 2017).

The dorsal caudate nucleus comprises 351 voxels in 2x2x2 mm MNI space, from which dopamine synthesis capacity values were extracted.

Figure S5. *Striatal subdivisions from Piray et al. (2017) including the dorsal caudate nucleus (green region) used in this study.”*

Prompted by the Reviewers’ questions, we also analyzed the relationship between model parameters and dopamine synthesis capacity in other striatal subregions. As expected, given that dopamine synthesis capacity values are highly correlated across the striatum, we also find a

significant correlation between our WM reliance parameter, ρ , and dopamine synthesis capacity in the posterior putamen ($r = 0.24, p = .029$) and a trending relationship in the medial caudate nucleus ($r = 0.18, p = .091$), but no other region (all other p 's > 0.21). Also, just like our dorsal caudate nucleus region, the RL learning rate is affected by a significant interaction between the dopamine synthesis capacity and methylphenidate in the posterior putamen ($\beta = .28; p = .046$) and the ventral striatum ($\beta = .28; p = .047$; all other p 's > 0.12).

Haber, S. N. & Knutson, B. The Reward Circuit: Linking Primate Anatomy and Human Imaging. *Neuropsychopharmacology* **35**, 4–26 (2009).

Badre, D. and Frank, M.J. (2012) Mechanisms of Hierarchical Reinforcement Learning in Cortico-Striatal Circuits 2: Evidence from fMRI. *Cerebral Cortex* **22**, 527–536

Fallon, S.J. *et al.* (2017) The Neurocognitive Cost of Enhancing Cognition with Methylphenidate: Improved Distractor Resistance but Impaired Updating. *J Cognitive Neurosci* **29**, 652–663

Piray, P., Ouden, H. E. M. den, Schaaf, M. E. van der, Toni, I. & Cools, R. Dopaminergic Modulation of the Functional Ventrodorsal Architecture of the Human Striatum. *Cereb Cortex* **27**, 485–495 (2017).

• Analyses of choice data was performed with brms using R. It would be informative to also report leave-one-out cross-validation via the loo package.

As recommended, we have used the loo package to perform leave-one-out cross-validation. For our full training model (of trial-wise logistic accuracy regressed on trial-level parameters and dopamine factors), all Pareto k estimates are good ($k < .5$), and the PSIS diagnostic plots revealed no data points with exaggerated influence on model parameters.

We did not compare across models in our analysis because our goal was not to identify new task variables which impact task performance. Instead, for our logistic regression, we adopted our analysis directly from a prior study (Collins *et al.*, 2017), and other studies that followed, including the exact set of predictors (delay, set size, and the number of previous correct iterations) and higher-order interactions previously shown to affect trial wise accuracy. In this study we also added dopamine variables (dopamine synthesis capacity and drug session) to test how dopamine would further interact with and modulate the influence of those task variables on accuracy.

Collins, A. G. E., Albrecht, M. A., Waltz, J. A., Gold, J. M. & Frank, M. J. Interactions Among Working Memory, Reinforcement Learning, and Effort in Value-Based Choice: A New Paradigm and Selective Deficits in Schizophrenia. *Biol Psychiat* **82**, 431–439 (2017).

• The authors conduct posterior predictive checks and parameter recoverability and meet high standards for computational modeling analyses. In the method section it is stated that a “wide range of models were fit via hierarchical maximum likelihood using the mfit toolbox”. Please provide more details what kind of different models these were and report details on model selection!

We thank the Reviewer for catching this oversight! We were unnecessarily vague in describing the models we fit. As we now explain in the methods on lines 700-705:

“We fit variants of the learning algorithms presented in (Collins and Frank, 2012; Master et al., 2020) striking a balance between the sensitivity of algorithms which allow working memory contributions to vary dynamically as a function of reliability and the simplicity of algorithms which maximize parameter recoverability. We settled on a new variant (described above) in which working memory reliance is a function of a baseline reliance parameter ρ , and the delay since the last correct iteration of a stimulus (Supplement).”

and in the Supplement, we articulate in greater detail in a Model Selection section:

“Model Selection

All the models we tested were variants of the original model proposed by Collins et al. (2012) and Master et al. (2020) combining contributions of both a WM and a RL module to action selection on every trial. A key difference regards mediation between the relative contributions of WM versus RL to choice on a particular trial. In the original Collins et al. model, mediation evolves dynamically as a function of the relative confidence ascribed to each module (so that RL contributes more with increasing experience, later in each block). In the Master et al. variant, the degree to which WM contributes to a choice is fixed across each block and varies, between blocks, as a function of set size. We initially adopted the Master et al. approach because it improved parameter recoverability yet found that, sensibly, we could not capture within-block changes in the effects of delay or the difference in set size effects, early versus late in a block.

Thus, as a compromise between these approaches, we developed a new subset of variants in which the relative contribution of the WM system was set for each block, as a function of set size, but also as a function of WM load relative to capacity. Specifically, we multiplied the WM reliance parameter (ρ) by the ratio of WM capacity to the number of unique intervening stimuli since the last correct response to a given item (see Equation 5). This subset of variants had improved recoverability and also captured within-block dynamics. We examined five closely related models in this subset:

Model A. A model in which we multiply ρ by $e^{((1-n_{delay})/10)}$ where n_{delay} is the total number of trials since the last correct iteration of a stimulus.

Model B. A model in which we multiply ρ by $\min\left(1, \frac{WM_{cap}}{n_{delay}}\right)$ where n_{delay} is the total number of trials since the last correct iteration of a stimulus

Model C. A model in which we multiply ρ by $\min\left(1, \frac{WM_{cap}}{n_{delay,k}}\right)$ where $n_{delay,k}$ is the number of unique, correct, intervening items since the last correct iteration of a stimulus.

Model D. A model in which we multiply ρ by $\min\left(1, \frac{WM_{cap}}{n_{delay,m}}\right)$ where $n_{delay,m}$ is the number of unique, intervening items since the last correct iteration of a stimulus.

Model E. A model in which we multiply ρ by $\min\left(1, \frac{WM_{cap}}{n_{delay,k}}\right)$ where $n_{delay,k}$ is the number of unique, intervening items since the last correct iteration of a stimulus. In this model we also allowed for low fidelity of the WM contents. Namely, the same noise parameter which was applied to choice during the test phase (Equation 7) was also applied to choice during the training phase. Note that we originally omitted this term from the training phase on the assumption that WM fidelity is high (when load is under capacity, during a given block). We considered adding the noise parameter back, in this model, because we wanted to know how it would interact with a dynamically varying contribution of working memory to choice, during training.

Although BIC scores were very similar across this subset of model variants, as shown in Figure S3, the winning model, Model C (BICc, in red) was slightly better for a majority of participants and drug sessions. In pairwise comparisons, Model C beat Model A for 93.1% of sessions, Model B for 87.3% of sessions, Model D for 52.2% of sessions, and Model E for 63.2% of sessions.

Figure S4. BIC scores for each participant in the placebo session, for five model variants.

• In the RL-WM task, steeper learning curves for smaller set sizes are thought to reflect

greater reliance on WM rather than faster RL. This is a plausible assumption and has been reasoned based on previous studies using similar task variants. However, could the authors provide evidence for this claim within the data, e.g. correlation with neuropsychological measures of WM such as digit span?

We think this is an excellent suggestion. As suggested, we tried two different types of tests of whether digit span is related to task performance which yielded suggestive but nonetheless inconclusive results. First, we fit a mixed-effects logistic regression of trial wise accuracy onto set size, previous iterations, total digit span, and their interactions, as predictors. This first analysis revealed that total digit span was a reliable predictor of (higher) accuracy ($p = .00843$, $b = .22$), controlling for other variables, and showed a trending interaction with previous iterations ($p = .066$, $b = .11$) such that people with higher digit span showed greater improvements in accuracy with each iteration, and a trending interaction with set size ($p = .074$, $b = .082$) such that people with higher span show less of an effect of set size. While this analysis supports the hypothesis that WM contributes to performance, it does not reveal anything selective about the contributions of WM to the learning process since greater performance could reflect greater reliance on WM at higher load levels, more effective WM across set sizes for those with higher span, or more (assuming shared variance between WM and RL systems) effective RL among those with higher span, at higher load levels.

In a complementary analysis, we extracted the random (subject-level) effects of our fully Bayesian logistic regression of trial-wise accuracy on delay, set size, and previous correct iterations, but without dopamine variables. Consistent with the hypothesis that set size and delay effects index WM contributions to the learning process, we find trending correlations between digit span and a smaller effect of set size on accuracy ($p = .069$, $r = .19$), a smaller effect of delay ($p = .069$, $r = .19$), and also a significant correlation between digit span and smaller set size by delay interaction term ($p = .038$, $r = .22$). Such correlations are consistent with the hypothesis that higher working memory span improves performance because it blunts the effects of set size and delay, and also blunts the effects of delay at higher set sizes. Here again, these results imply a role for WM, but it is unclear whether people with higher span show less of an effect of delay at higher set size because their WM capacity is more robust across load levels, or because they don't down-regulate WM contributions with load, to the same degree as people with lower span.

These results provide indirect support for the hypothesis that higher WM capacity (as indexed by digit span) relates to faster learning as a function of set size, yet inferences are indeterminate. Part of the indeterminacy is that digit span is a between subjects' measure whereas the core question is a within subjects' question: what happens as load level changes? As a result, it is unclear what kinds of relationships we should expect. Do people who have higher span rely on WM more overall? Do they rely on WM relatively more or less as load decreases? In our results, we find evidence that people with higher span have less load-related effects. This could reflect that they are better at WM tasks, regardless of load, or that they rely on WM relatively less as set size increases. As such, given the indeterminacy of the evidence, we have ultimately decided not to include these results in the manuscript. We can certainly reconsider (or consider other kinds of tests) if the Reviewer feels it is warranted.

We note that beyond these analyses, results reported in prior studies are consistent with the Reviewer's intuition. For example, Collins et al. (2014) found that neuropsychological measures of working memory ability correlated with model-based parameters related to working memory

contributions to task performance including working memory reliability and working memory decay. Similarly, as noted above in our response to Reviewer 2, Collins & Frank (2018) and Rac-Lubashevsky et al. (2023) both found neural signatures of WM engagement, in trial-wise EEG dynamics, which were tradeoff with neural signatures of RL engagement – and this tradeoff varies by set size. In sum, these neural data further support the hypothesis that WM contributes to faster stimulus-response acquisitions, especially in smaller set size blocks.

Collins, A. G. E. & Frank, M. J. Within- and across-trial dynamics of human EEG reveal cooperative interplay between reinforcement learning and working memory. *Proceedings of the National Academy of Sciences* **115**, 2502–2507 (2018).

Rac-Lubashevsky, R., Cremer, A., Collins, A.G.E., Frank, M.J.*. & Schwabe, L.*. (2023). Neural index of reinforcement learning predicts improved stimulus-response retention under high working memory load. *Journal of Neuroscience*, *43*(17), 3131–3143.

Collins, A. G. E., Brown, J. K., Gold, J. M., Waltz, J. A. & Frank, M. J. Working Memory Contributions to Reinforcement Learning Impairments in Schizophrenia. *J Neurosci* **34**, 13747–13756 (2014).

• It would be helpful to publish the analyses scripts together with the paper including the R scripts for full transparency.

We agree wholeheartedly. For our Reviewers, we have made the data and the scripts necessary to perform the logistic regressions, to fit the models, and to perform statistical analyses on the Donders Data Repository at:

<https://data.ru.nl/login/reviewer-2714769840/H5IOOB4CVHHDKZ4MAHD3VPZMRREQQW26U7IETOQ>

If the manuscript is accepted, we will make this Repository publicly available.

Reviewer #1 (Remarks to the Author):

The authors have done a good job improving clarity and transparency of the manuscript and they have addressed most of my comments. There are, however, some remaining points:

The cost interpretation of the fact that participants under methylphenidate choose the most rewarding choice more frequently than under placebo when the set size difference is unfavorable to the most rewarding option is straight forward. However, I am still not completely convinced that this interpretation is so straight forward to the fact that participants choose the most rewarding option less under methylphenidate than placebo when it comes from smaller set sizes. Why would they discount the value of those options? As the points in figure 6B and C involve choices with different discriminability of value, it may be that participants under methylphenidate are better at learning the value of all the options, also those of the less rewarding option coming from the bigger set size. This may result in less subjective value difference and less probability of choosing the most rewarding one. In order to reject this explanation, the authors would need to show that there are no difference between methylphenidate and placebo in discriminability, that is on a plot like the one displayed in figure 6A.

We thank the Reviewers for their thoughts on this point. We had also considered the possibility that the drug effect here might have something to do with overall higher accuracy, during the test phase, of MPH versus PBO. However, we can confirm that there is in fact neither a difference in terms of test phase accuracy on the MPH vs PBO condition, nor a drug by Q value difference interaction (both p 's > 0.57).

We believe that participants choose the more rewarded option at lower rates on MPH versus PBO when the set size difference is negative (i.e., the more rewarded option comes from a smaller set size block) because, on PBO, there is an implicit effort cost discounting embedded in the learned value of the less rewarded option (b.c. it comes from a larger set size block) and this effect is attenuated on MPH. That is, MPH blunts the cost discounting effect which would otherwise lead to higher selection rates of the more rewarded option, on trials when the more rewarded option comes from a smaller set size block.

In any case, I believe that the authors make too strong claims in the title, the abstract and conclusion about the effects of methylphenidate on effort costs considering that the evidence provided in the data is only indirect. At the least, the authors should be very clear to the reader that the data does not have any explicit measure of effort costs and that the evidence supporting their claim in the data is at least indirect and based on interpretation of previous studies.

We are sympathetic to the Reviewer's perspective here. Indeed, the measure is not an explicit cost measure, but rather implicit in the sense that it is inferred from the effect of putative costs on choice behavior. We note, however, that this inferential logic is also used in a closely-related class of revealed preference studies which also infer either costs or benefits from their respective effects on choice behavior. Revealed preference elicitation, where participants are asked to choose, e.g., between a more demanding task for more reward or a less demanding task for less reward (including our own prior work: Westbrook et al., 2020) rely on implicit assumptions that the options participants choose, when allowed to pick whichever they prefer, reflect underlying cost and benefit factors, even when participants are not explicitly instructed to weigh either. Although we did not ask participants which stimulus they preferred, but rather which was rewarded more, our inference may be considered even stronger than if we had used a revealed preference approach. Namely, our method avoids concerns about perceived risks confounding our effort cost interpretation. Unlike a method where we ask participants which option they prefer, we don't have to worry that they might avoid choosing a more demanding option because it is associated with higher risk. Thus, the adverse effect of set size differences on the perception that a stimulus was rewarded more can be more cleanly interpreted as an implicit effort cost effect, not an implicit risk effect.

Nevertheless, we agree that it is appropriate to be clear that our inference assumes implicit effects of costs on choice behavior. As such, we have amended our description in the Abstract and Discussion by using the phrase "implicit effort sensitivity" and also modified language to observe the nature of the finding more strictly (that set size differences affect the perceived rate at which stimuli were rewarded), while separately indicating that we infer an *implicit* effort cost effect from this pattern.

Lines 27 – 32:

“Methylphenidate also blunts implicit effort cost sensitivity. Computational modeling reveals that individuals with high dopamine synthesis rely more on WM, while methylphenidate boosts their RL rates. The D2 receptor antagonist sulpiride reduces accuracy due to diminished WM involvement and faster WM decay. We conclude that dopamine enhances both slow RL, and fast WM, by promoting plasticity and reducing implicit effort sensitivity.”

Line 311:

“Methylphenidate blunts implicit effort cost sensitivity”

Lines 326 – 335:

“Also replicating prior work [39], we find that cached reward values are perceived as less rewarding when they came from larger set-size blocks – an effect previously interpreted as reflecting implicit sensitivity to effort costs. Specifically, controlling for the rewards people received for each stimulus during the learning phase, participants assign a lower reward value to stimuli that had been encountered in larger set size blocks. This effect is captured by a negative effect of the difference in set sizes between the stimulus which objectively received more rewards and the stimulus which received fewer rewards ($\beta = -.28$; $p = 7.1 \times 10^{-8}$; Figure 6B: collapsed across differences in actual reward rates). That is, while participants track the rewards associated with each stimulus, they treat rewards received in the context of higher set sizes, and thus higher WM demands, as subjectively less rewarding.”

Lines 359 – 372:

“Importantly, we also find evidence that striatal dopamine blunts this implicit effort discounting effect. Specifically, the effect of set size on perceived rewards was significantly less on methylphenidate versus placebo ($\beta = .12$; $p = .025$; Figure 6B). This effect of set size on perceived rewards was not influenced by dopamine synthesis capacity ($\beta = -.01$; $p = .85$). This effect also does not reflect differences in the ability of participants to track reward statistics between drug sessions. Indeed, test phase accuracy (the rate at which participants correctly identify the more rewarded stimulus) is no different on methylphenidate versus placebo, and there is also no drug by reward rate difference interaction (both p 's > 0.57). In a different experiment conducted within the current study we find that striatal dopamine signaling increases sensitivity to reward benefits and decreases sensitivity to effort costs during decision-making about cognitive effort [24]. This result converges with the present finding that dopamine signaling can blunt implicit effort sensitivity during reward learning. Thus, by amplifying striatal dopamine signaling, methylphenidate not only increases sensitivity to benefits versus costs during action selection [24], but may also alter how people learn about effort costs and benefits in the first place.”

Lines 420 – 431:

“The present findings constitute an important complement to that conclusion. Namely, while the prior study primarily examined prospective decisions about WM tasks, after effort costs had already been learned, the present study reveals that dopamine may shape value learning about effort costs themselves. Specifically, we find that people treat rewards earned in the context of higher WM demands as subjectively less rewarding – an effect which we interpret as implicit discounting rewards by increasing cognitive effort costs. Critically, we find that methylphenidate blunts this implicit effort-discounting effect. Thus, stronger striatal dopamine signaling may make people both more willing to exert effort for tasks they have experienced in the past and experience the tasks as less costly when they learn about them in the first place (cf. [50]). An important caveat is that, unlike prior work showing that methylphenidate can alter explicit cost-benefit decision-making, our measure of effort costs is implicit and the inference that dopamine modulates cost learning during task performance is indirect.”

Lines 537 – 539:

“Finally, we find evidence that pharmacological enhancement of striatal dopamine signaling can blunt implicit effort cost learning that happens when people perform demanding tasks.”

On lines 187-193, it is not clear whether the results are corrected or uncorrected. The text is easily read as if they were corrected.

We thank the Reviewer for helping us recognize the lack of clarity here. To simplify things, we took out the simple correlations for dopamine synthesis capacity with reliance on WM, separately, in the placebo and sulpiride sessions, and replaced those with an overall ANOVA which confirms a main effect of dopamine synthesis capacity on WM reliance, across all drug sessions, even after applying Bonferroni correction.

Lines 203 – 215

“A hierarchical regression of the parameter ρ (WM reliance) on dorsal caudate dopamine synthesis capacity and drug, controlling for session number, reveals a positive effect of dopamine synthesis capacity on placebo ($\beta = .23$; $p = .029$; Figure 4A), no effect of methylphenidate ($\beta = .083$; $p = .48$), and a negative effect of sulpiride ($\beta = -.35$; $p = .0028$; Figure 4B; Supplemental Table S3) on WM reliance. We note that we separately regressed multiple model parameters onto dopamine factors and this dopamine synthesis capacity effect does not survive when we correct for these multiple comparisons ($p_{\text{Bonferroni}} = .087$; Supplemental Tables S3–S5). Nevertheless, a two-way ANOVA, correcting for multiple comparisons, confirms a main effect of dopamine synthesis capacity ($F(1,251) = 8.94$, $p_{\text{Bonferroni}} = .0093$) across sessions and a main effect of drug across participants ($F(2,251) = 4.44$, $p_{\text{Bonferroni}} = .039$), but no dopamine synthesis capacity by drug interaction ($F(2,251) = 2.06$, $p_{\text{Bonferroni}} = .39$). Collectively, these results support the hypotheses that people who can synthesize dopamine at a higher rate rely more on WM in general and that sulpiride reduced WM reliance.”

Finally, it is comforting to see that the removal of outliers does not fully drive the reported correlations between synthesis capacity and model parameters. However, considering that the exclusion criteria is rather harsh (2 SD instead of the more standard 3SD) and implies exclusion of a substantial number of participants (5%), I would suggest that the results including the outliers are reported in the supplement. In this way, the reader will be aware of the impact of outlier removal. An alternative to avoid exclusion of so many participants would be to normalize the data.

As the Reviewer requests, we have now added details about the effects of including or excluding outlier parameter values in the Supplement, immediately following the histograms which show the distribution of parameter values and where the cutoffs lie.

“Figure S3. *The distribution of fitted parameter values in each drug session. Vertical red lines indicate outlier values based on a +/- 2 SD cutoff.*

The removal of outliers did not affect any of the core inferences in the manuscript. There are some changes in exact p-values when we include sessions with outlier values for two

parameters (WM decay ϕ , and WM capacity WM_{cap}). However, in neither case do the inferences which we make change fundamentally.

Regarding the WM decay term (ϕ), if we do not remove sessions with outlier value estimates, the overall positive relationship between DA synthesis capacity and WM reliance (ρ) across sessions is no longer significant ($\beta = .16$; $p = .10$). However, positive correlations remain between DA synthesis capacity and WM reliance in the placebo ($r = .24$; $p = .028$) and sulpiride sessions ($r = .28$; $p = .0098$). Similarly, the relationship between dopamine synthesis capacity and the learning rate is no longer significant in the methylphenidate session ($\beta = .17$; $p = .13$). However, a two-way interaction between dopamine synthesis capacity and methylphenidate remains, indicating that the relationship between the learning rate and dopamine synthesis capacity is stronger on methylphenidate versus placebo ($\beta = .28$; $p = .032$). Thus, in both cases, including outlier WM decay parameter values does not fundamentally alter the interpretation, even if there are small changes to the resulting p-values.

When we do not remove outlying WM capacity values (i.e., $WM_{cap} = 2$), the overall positive relationship between DA synthesis capacity and WM reliance across sessions is no longer significant ($\beta = .19$; $p = .062$). However, positive correlations remain between DA synthesis capacity and WM reliance in the placebo ($r = .22$; $p = .041$) and sulpiride sessions ($r = .28$; $p = .0098$) separately, indicating that not excluding on the basis of outlying WM capacity also does not fundamentally alter these inferences.”

Reviewer #2 (Remarks to the Author):

The authors addressed all my concerns and the manuscript has improved significantly.

We are glad that we were able to address the Reviewer’s concerns satisfactorily.